# How well do mothers recall their own and their infants' perinatal events? A two-district study using cross-sectional stratified random sampling in Bihar, India

Joseph James Valadez  ,[1] Baburam Devkota,[1] Caroline Jeffery,[1] Wilbur C Hadden[2]

[1]Department of International Public Health, Liverpool School of Tropical Medicine, Liverpool, UK
[2]Department of Sociology, University of Maryland at College Park, College Park, Maryland, USA

**Correspondence to**
Dr Joseph James Valadez;
joseph.valadez@lstmed.ac.uk

## ABSTRACT

**Objective** Global monitoring of maternal, newborn and child health (MNCH) programmes use self-reported data subject to recall error which may lead to incorrect decisions for improving health services and wasted resources. To minimise this risk, samples of mothers of infants aged 0–2 and 3–5 months are sometimes used. We test whether a single sample of mothers of infants aged 0–5 months provides the same information.

**Design** An annual MNCH household survey in two districts of Bihar, India (n=6 million).

**Participants** Independent samples (n=475 each) of mothers of infants aged 0–5, 0–2 and 3–5 months.

**Outcome measures** Main analyses compare responses from the samples of infants aged 0–5 and 0–2 months with Mantel-Haenszel-Cochran statistics using 51 indicators in two districts.

**Results** No measurable differences are detected in 79.4% (81/102) comparisons; 20.6% (21/102) display differences for the main comparison. Subanalyses produce similar results. A difference detected for exclusive breast feeding is due to premature complementary feeding by older infants. Measurable differences are detected in 33% (8/24) of the indicators on Front Line Worker (FLW) support, 26.9% (7/26) of indicators of birth preparedness and place of birth and attendant, and 9.5% (4/42) of the indicators on neonatal and antenatal care.

**Conclusions** Differences in FLW visits and compliance with their advice may be due to seasonal effects: mothers of older infants aged 3–5 months were pregnant during the dry season; mothers of infants aged 0–2 months were pregnant during the monsoons, making transportation difficult. Useful coverage estimates can be obtained by sampling mothers with infants aged 0–5 months as with two samples suggesting that mothers of young infants recall their own perinatal events and those of their children. For some indicators (eg, exclusive breast feeding), it may be necessary to adjust targets. Excessive stratification wastes resources, does not improve the quality of information and increases the burden placed on data collectors and communities which can increase non-sampling error.

## Strengths and limitations of this study

► The data were produced using stratified random sampling with no apparent design effect leading to an efficient use of information.
► Data were collected from female participants by female data collectors which is likely to have reduced non-sampling error.
► The large study population covers a large geographical area, reducing the likelihood that the results are pertinent only to a small group of mothers with infants and may be generalisable.
► Both weighted and unweighted results are presented giving strength to the conclusions.
► Due to insufficient overlap of variables in the 0–5 months' sample and the 3–5 months' sample, comparison between the 3–5 and the 0–5 months' sample was not possible.

## INTRODUCTION

The progress towards United Nations' Sustainable Development Goal (SDG) 3 is measured with nine targets, including the maternal mortality ratio (MMR) and the under-five mortality rate (U5MR).[1 2] In India, Bihar is one of the largest (population 110 million) and poorest (53% of households are in the lowest wealth index quintile of India[3]) states with high child and maternal mortality (U5MR=54, MMR=208),[4] and is a priority for donor support for health systems strengthening (see the study by Karvande *et al*[5] for an evaluation of the healthcare system in Bihar).

To accelerate progress towards achieving SDG 3, state governments in India pursue programmes of community-based care (see the studies by Mohan *et al* and Neogi *et al*[6 7] for descriptions and assessments of this approach). Since 2011, the Bihar Ministry of Health has supported an Integrated Family

Health Initiative to improve the availability, quality and use of prenatal, perinatal and postnatal care for mothers and infants.[8]

The usual way to monitor progress towards achieving these goals is with household surveys. Perhaps the most commonly used surveys are cluster sample surveys such as the Demographic and Health Surveys and the Multiple Indicator Cluster Surveys.[9 10] An alternative design is Lot Quality Assurance Sampling (LQAS) which provides comparable data but is decentralised to local health services organisations and more useful for management and programme planning.[11] Several states in India find it benefits their programmes.[12] Surveys rely on the reports of mothers of infants and young children, but these reports are subject to several sources of potential error and bias through interviewees not knowing, forgetting and having memory errors.[13 14] Studies have shown both that mothers can accurately report significant facts about the birth and care of their children many years after the event,[15] but also that even immediately after giving birth mothers may misreport details.[16–18] Studies of mothers recall of their children's vaccination status concluded that due to offsetting errors of maternal reports, the resulting data accurately measured vaccination rates[19]; the pattern of error revealed that mothers whose children are up-to-date or nearly so tended to underestimate their child's vaccination status while mothers whose children have few vaccinations overestimate their coverage.

To improve the validity of collected data, knowledge, practice and coverage, surveys have used samples of mothers of infants aged 0–11 months or 0–5 months and children aged 6–11 months. In Bihar, local organisations departed from this convention of sampling among these three cohorts of children under 1 year of age and have been monitoring their programmes' progress by sampling five dedicated cohorts: mothers of children aged 0–2, 3–5, 6–8, 9–11 and 12–23 months with indicators focused on antenatal care, safe delivery practices, infant and young child-feeding practices, immunisation, treatment seeking and more. To avoid the possibility of maternal recall error, each of the five cohorts was asked questions particularly relevant to a child's specific age group.

In countries such as India with high maternal and child mortality rates, regular monitoring of related health service coverage is critical to reducing these rates. However, survey designs should be affordable and sustainable for local health systems; they should also produce precise, unbiased estimates.[20] In this study, we explore whether information is gained by sampling cohorts of children aged 0–2 and 3–5 months or whether sample sizes can be reduced by 50% by creating one sample cohort aged 0–5 months.

The research question we address is: 'Do the health service delivery coverage estimates from a sample of mothers of infants aged 0–5 months differ from those obtained from a sample of mothers of infants ages 0–2 months?' A corollary to this question is: 'Do mothers of infants 3–5 months of age display more recall error relative to mothers of infants 0–2 months of age for antenatal, delivery or young infant health practices?' We compare district coverage estimates obtained from two independent samples of infants aged 0–2 months and 0–5 months. The implications of this study are important for health systems researchers needing results to appraise and improve their programmes.

## METHODS

To answer this question, we collected information from a sample of mothers with infants aged 0–5 months and a sample of mothers with infants aged 0–2 months in two districts. This study took place within the context of a larger survey that also sampled children aged 3–5, 6–8, 9–11 and 12–23 months. These four latter samples used questionnaires with variables that either did not overlap at all or overlapped on very few indicators with the questionnaires used to interview the 0–5 and 0–2 months' samples of infants. Due to this constraint, in this study, we only use the two aforementioned groups to assess the measurement of the indicators and refer only to them for the remainder of this paper. The household sampling design we used is a stratified random sample.[21] Within each district, the strata are administrative units of the health system which in Bihar is called a *block*. Within each block, the primary sampling unit is the Anganwadi Centre (Community Health Subcentre) Catchment Area (ACCA); 19 ACCAs are selected from each block with probability proportional to size. From each ACCA, one respondent is randomly selected from each age group under study using segmentation sampling.[22 23] The sample of 19 mothers in each block is chosen to maximise the probability of correctly classifying a block with reference to performance targets on health-related indicators (95% reliability) while balancing the probability (10% margin of error) of incorrectly classifying a block and thereby failing to recognise either the accomplishments of local healthcare delivery systems or the local population's healthcare needs.[22] For this purpose, principles of LQAS were used along with established probability tables.[24–26]

There are 14 and 11 blocks in Gopalganj and Aurangabad (n=6 million), the two districts selected for this study, respectively. The total sample sizes are: (a) Gopalganj: 19×14 blocks=266 infants aged 0–2 months and 266 infants aged 0–5 months, and (b) Aurangabad: 19×11 blocks=209 infants aged 0–2 months and 209 infants aged 0–5 months. The 0–5 months old sample is distributed as 60% 0–2 months old and 40% 3–5 months old.

Using summary data from each of the two samples, we analyse the data with Cochran-Mantel-Haenszel (CMH)[27] tests for 51 dichotomous indicators (online supplementary table S1) common to the two samples. The CMH tests theoretically have a $\chi^2$ probability distribution with 1 df. With a sufficient number of respondents or a sufficient number of blocks, the CMH test is equivalent to a conditional logistic regression (Agresti, pp114–115[28]). In this analysis, both the number of respondents and the number

**Table 1** Number of indicators by probability of a difference between the 0–2 and 0–5 months' samples for weighted and unweighted samples

| | Weighted | | | | | |
| | Aurangabad | | Gopalganj | | Total | |
| Unweighted | ≥0.05 | <0.05 | ≥0.05 | <0.05 | ≥0.05 | <0.05 |
|---|---|---|---|---|---|---|
| ≥0.05 | 40 | 3 | 41 | 4 | 81 | 7 |
| <0.05 | 0 | 8 | 1 | 5 | 1 | 13 |

of blocks only approach sufficiency. Consequently, the calculated $\chi^2$ and probabilities must be considered as approximations of their true values.

We calculate both unweighted and weighted estimates. The unweighted estimates permit the results from smaller blocks to have equal weight vis à vis larger ones. Since the research question concerns an analysis of which age cohort is most informative, the weighted estimates may not be as useful as the unweighted ones. However, the weighted estimates provide better point estimates of the indicators at the district level. The effect of the weights on the $\chi^2$ statistics is to increase the contribution of the larger blocks and decrease the contribution of the smaller blocks. Hence, we report both sets of results (online supplementary tables S2-S3).

The $\chi^2$ probability distribution puts the differences between the districts on a probability scale (online supplementary table S2). To determine meaningful differences in responses between the two age cohorts, we used a probability of 0.05 as a cut-off value and considered differences with probabilities less than 0.05 to be possibly meaningful and those with larger probabilities to be likely due to sampling errors. With 102 comparisons (51 indicators weighted or unweighted), we must expect some to exceed this cut-off by chance alone. If all of the comparisons were independent, we might randomly find five differences,

but many of the indicators measure related events (eg, number of ANC visits and tetanus toxoid vaccinations) and the weighted and unweighted estimates were similar, so these indicators were not all independent, and it was not possible to calculate an expected number of differences nor was it appropriate to interpret these probabilities as measures of 'statistical significance'.

### Patient and public involvement

This study does not involve patients. Also, the public was not involved in the design, conduct and reporting of the research. The public was engaged as interviewees. To ensure local engagement, we coordinated with the Bihar Ministry of Health, local implementing non-governmental organisations and our donor. We also shared the results with them and offered further dissemination of results.

### RESULTS

We find a high level of agreement between the two samples (table 1). Out of 102 weighted and unweighted comparisons between the estimates from the 0–2 and 0–5 months' samples, there is no probable difference in 81 (79.4%) in both the unweighted and weighted estimates. We detect that probable differences exist for 13 comparisons (12.7%). For the remaining eight comparisons, the weighted and unweighted estimates disagree. The weighted estimates find seven differences that the unweighted estimates do not; the unweighted estimates find one difference that the weighted estimates do not find.

For different health service domains, the number of indicator comparisons varies from two (exclusive breast feeding; EBF) to 24 concerning home visits by Front Line Worker (FLW) support (table 2). The two principal FLW are Anganwadi workers and Accredited Social Health Activists (ASHA).

In the FLW support domain, 33% of comparisons have probable differences. The neonatal health domain has 20

**Table 2** Number of indicator comparisons by subject domain showing a measurable difference using weighted and unweighted estimates of 0–2 and 0–5 months' samples

| Health service domain | Total comparisons | No measurable difference between 0–2 and 0–5 months' results | Measurable difference between 0–2 and 0–5 months' results | | | Per cent indicators with different results |
| | | | Both | Unweighted only | Weighted only | |
|---|---|---|---|---|---|---|
| Antenatal care | 22 | 21 | 0 | 0 | 1 | 5 |
| Place of birth and attendant | 8 | 6 | 1 | 0 | 1 | 25 |
| Birth preparedness | 18 | 13 | 3 | 0 | 2 | 28 |
| Front Line Worker support | 24 | 16 | 6 | 0 | 2 | 33 |
| Maternal health | 8 | 8 | 0 | 0 | 0 | 0 |
| Neonatal care | 20 | 17 | 1 | 1 | 1 | 15 |
| Exclusive breast feeding | 2 | 0 | 2 | 0 | 0 | 100 |
| Totals | 102 | 81 | 13 | 1 | 7 | 21.9 |

comparisons and the birth preparedness domain has 18; in these domains 15% and 28% show probable differences, respectively. The place of birth and attendant domain, and maternal health domain each have eight comparisons with 25%, or two comparisons, and 0 comparisons, respectively, showing a possible difference. The differences between the two samples cluster around home visits from FLW and behaviours associated with birth preparedness and neonatal care. Details of these differences are listed in table 3.

For two indicators, both the weighted and unweighted estimates display probable differences between the 0–2 and 0–5 months samples in both districts. For indicator #52, the proportion exclusively breast feeding, the 0–5 months' cohort has the lower estimate, and indicator #24, the proportion of mothers visited by an ASHA at least once during their last pregnancy, the 0–5 months' sample gives the higher estimate, about 74%, compared with 63% in the 0–2 months' sample (online supplementary tables S2-3).

Additional analyses comparing subsamples of mothers of infants aged 0–2 months and 3–5 months from the 0–5 months' sample, the sample of mothers of infants aged 0–2 months and the subsample of infants aged 0–2 months, and the sample of mothers of infants aged 3–5 months and the subsample of infants aged 3–5 months produced similar results (online supplementary text, tables S4a-b, S5a-b and S6).

## DISCUSSION
### Statement of principal findings
There are no measurable differences in coverage estimates for 79.4% (81 comparisons) of the indicator comparisons between the samples of mothers with infants aged 0–2 months versus mothers of infants aged 0–5 months; 12.7% (13 comparisons) display measurable differences. The remaining 7.8% (eight comparisons) display discrepancies between the weighted and unweighted estimates.

### Strengths and weaknesses of the study
The strengths of this study are that it compares estimates from two independent samples and that there are many estimates from diverse domains. The weaknesses of this study are that the data have been collected in only two districts of one state in India and in different months of a single year, and that indicators from the sample of mothers of 3–5 months old infants comparable to those of the 0–2 months old infants, using the same questionnaire, have not been collected. Supplemental analyses comparing 0–2 and 3–5 months subsamples of the 0–5 sample did not uncover evidence of bias due to the combination of these two age groups.

### Strengths and weaknesses in relation to other studies
Other studies of maternal recall bias have sought a 'gold standard' to represent reality and to evaluate measures. Our study, of course, is interested in reality, but this study compares alternative measures needed to assess the Bihar health programme. It also uses a complete sample of the age grouping under study rather than just a sub-sample of

a larger age grouping. A weakness of this approach is that the analysis does not result in a formal statistical test; our conclusion is based on the weight of the evidence.

### Meaning of the study
The evidence indicates that samples of the broader group yield comparable results to those of the narrower age group. It is not necessary to double the total sample by measuring independently 0–2 months' and 3–5 months' cohorts of children. These results also tend to dispel the hypothesis that maternal recall is problematic for mothers during the first 6 months following delivery. Our results are more consistent with conclusions presented in earlier research,[15] and they support those organisations collecting data with 0–5 months' cohorts.

Indicator #52, EBF, displays two comparisons measuring decreases in both districts. This is not surprising as fewer infants are expected to be exclusively breast fed in a sample ranging from 0 to 5 months than a sample ranging 0–2 months; mothers introduce complementary feeding and liquids as infants age despite this being a health risk. This difference could be accommodated by adjusting expectations and targets for the indicator.

### Unanswered questions and future research
Further investigation and consideration of the differences is warranted. The eight differences found in the FLW support indicators deserve more scrutiny. Seven show higher estimates for the 0–5 months' cohort, and one has a higher estimate for the 0–2 months' cohort. The former seven differences may be due to excessive rainfall during July–September (monthly 2016 average 288 mm, range: 151–35 mm) versus the lesser rainfall during October–June (monthly 2016 average 33 mm, range: 0.0–129 mm) which in the last trimester may have reduced the access of ASHA in the 0–2 months' cohort.[29] Indicators such as these may be particularly sensitive to rainfall and may explain why more mothers in the 0–5 months' cohort displayed higher FLW visitation estimates since FLW were not impeded by the monsoon and the resulting muddy roads.

Differences in birth preparedness and institutional birth may be a consequence of differences in rainfall or in FLW support; the results signal a need for more careful planning when transportation is difficult and decreases the effectiveness of FLW by reducing their access to women. Or, some of these differences may just be due to noise in the data.

## CONCLUSIONS
Overall, the answer to the research question, 'Can one get the same district coverage estimates from a sample of mothers of infants aged 0–5 months as from a sample of mothers of infants aged 0–2 months?' is yes. This result can be paraphrased as: mothers do not display increased recall errors of their perinatal healthcare behaviour in a cohort of mothers with infants aged 0–5 months as

**Table 3** Indicators by health service domain showing measurement differences between 0–2 and 0–5 months' samples

| Health service domain and indicator | Indicator no | District | Weighted coverage (%) | | P value | | Estimate type |
|---|---|---|---|---|---|---|---|
| | | | 0–2 months | 0–5 months | Unweighted estimate | Weighted estimate | |
| Antenatal care | | | | | | | |
| Proportion of mothers of infants (0–2/0–5 months) registered during their last pregnancy | 1 | Aurangabad | 85.2 | 77.6 | 0.0552 | 0.0365 | Weighted |
| Place of birth and attendant | | | | | | | |
| Proportion of mothers of infants (0–2/0–5 months) whose last child was delivered at a public facility | 38 | Gopalganj | 51.2 | 61.6 | 0.0363 | 0.0159 | Both |
| Proportion of mothers of infants (0–2/0–5 months) whose last child was delivered at a health facility (private or public facility) | 37 | Gopalganj | 78.9 | 85.5 | 0.0643 | 0.0459 | Weighted |
| Birth preparedness | | | | | | | |
| Proportion of mothers of infants (0–2/0–5 months) who planned transportation to health facility in their last pregnancy (home and institutional delivery) | 15 | Gopalganj | 45.7 | 56.0 | 0.0266 | 0.0158 | Both |
| Proportion of mothers of infants (0–2/0–5 months) who identified persons to care for the baby immediately after birth (home+institutional delivery) | 17 | Gopalganj | 51.8 | 62.6 | 0.0255 | 0.0103 | Both |
| Proportion of mothers of infants (0–2/0–5 months) who planned for institutional delivery and identified person to accompany her during the delivery | 23 | Aurangabad | 62.5 | 47.0 | 0.0039 | 0.0052 | Both |
| Proportion of mothers who planned for institutional delivery of infants (0–2/0–5 months) who had a new blade and thread for their delivery | 19 | Aurangabad | 23.5 | 14.5 | 0.062 | 0.0429 | Weighted |
| Proportion of mothers who planned institutional delivery of infants (0–2/0–5 months) who arranged clean cloth for mothers and baby | 21 | Aurangabad | 43.6 | 31.2 | 0.0546 | 0.0137 | Weighted |
| FLW support | | | | | | | |
| Proportion of mothers of infants (0–2/0–5 months) who were visited by ASHA at least once during their last pregnancy | 24 | Aurangabad | 62.2 | 75.2 | 0.0023 | 0.0042 | Both |
| | | Gopalganj | 63.5 | 73.0 | 0.0284 | 0.0175 | Both |
| Proportion of mothers of infants (0–2/0–5 months) visited at home by FLWs at least once during their last pregnancy | 26 | Aurangabad | 63.5 | 76.7 | 0.0021 | 0.0032 | Both |
| Proportion of mothers of infants (0–2/0–5 months) visited at home by ASHA within 24 hours of last delivery | 31 | Aurangabad | 29.9 | 44.9 | 0.0009 | 0.0016 | Both |
| Proportion of mothers of infants (0–2/0–5 months) visited at home by any FLW within 24 hours of last delivery | 33 | Aurangabad | 32.2 | 46.7 | 0.0015 | 0.0026 | Both |
| Proportion of mothers of infants (0–2/0–5 months) visited at home by any FLW within first week of last delivery | 35 | Aurangabad | 44.5 | 59.3 | 0.0018 | 0.0026 | Both |
| Proportion of mothers of infants (0–2/0–5 months) visited at home by any Anganwadi worker (AWW) within the first week of the last delivery | 34 | Gopalganj | 14.4 | 8.9 | 0.0959 | 0.0471 | Weighted |
| Proportion of mothers of infants (0–2/0–5 months) visited by ASHAs at least once during their last trimester of pregnancy | 27 | Gopalganj | 52.6 | 61.1 | 0.0617 | 0.0449 | Weighted |
| Infant care | | | | | | | |
| Proportion of infants aged 0–2/0–5 months who were delivered at home continued with dry cord care | 51 | Aurangabad | 78.0 | 45.4 | 0.0001 | 0.0006 | Both |
| | | Gopalganj | 63.7 | 41.1 | 0.0431 | 0.0627 | Unweighted |
| Proportion of infants aged 0–2/0–5 months weighed after birth (public facility/private facility/home) | 48 | Gopalganj | 70.7 | 78.2 | 0.0727 | 0.0464 | Weighted |

Continued

## Table 3   Continued

| Health service domain and indicator | Indicator no | District | Weighted coverage (%) | | P value | | Estimate type |
| --- | --- | --- | --- | --- | --- | --- | --- |
| | | | 0–2 months | 0–5 months | Unweighted estimate | Weighted estimate | |
| **Exclusive breast feeding** | | | | | | | |
| Proportion of infants (0–2/0–5 months) breast fed in the past 24 hours (exclusively breast fed) | 52 | Aurangabad | 69.2 | 59.7 | 0.0229 | 0.0411 | Both |
| | | Gopalganj | 82.1 | 68.4 | 0.0001 | 0.0003 | Both |

ASHA, Accredited Social Health Activist; FLW, Front Line Worker.

compared with mothers with younger infants. Substantial resources and effort can be saved using a survey design that avoids needless expenses to collect data that provides insubstantial amounts of information. It also reduces the burden on data collectors and community participants. Fatigue to both groups can result in needless non-sampling error.

**Acknowledgements**  We gratefully acknowledge the essential roles of Hemant Das, Alok Prahdan and Sanjay Biswa for their careful field work supporting the LQAS survey and the HMIS data retrieval. We thank Professor Imelda Bates, Professor Brian Faragher and Nancy Vollmer for their valuable feedback on an earlier version of this manuscript.

**Contributors**  JJV, BD developed the research question and survey design; WCH, CJ carried out the statistical analyses; JJV obtained the funding and donor support for the research; BD trained and managed the survey teams in Bihar; JJV, WCH, CJ interpreted the data; CJ responsible for data curation; all authors wrote and reviewed the paper.

**Funding**  This research was funded by the Bill and Melinda Gates Foundation Investment ID OPP1142889

**Competing interests**  None declared.

**Patient consent for publication**  Not required.

**Ethics approval**  The Ethical Committees of the Indian Institute of Public Health (No IIPHB-IEC-2016/010) and the Liverpool School of Tropical Medicine Research Ethics Committee approved the protocol, study instruments and consent procedures for the data collection of the household surveys (Research Protocol 16-023).

**Provenance and peer review**  Not commissioned; externally peer reviewed.

**Data availability statement**  Data are available upon reasonable request.

**ORCID iD**
Joseph James Valadez http://orcid.org/0000-0002-6575-6592

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
