## [Reviewer comments · BMJ Open]

ARTICLE DETAILS

TITLE (PROVISIONAL)	How well do mothers recall their own and their infants' perinatal events? A two-district study using cross-sectional stratified random sampling in Bihar, India
AUTHORS	Valadez, Joseph; Devkota, Baburam; Jeffery, Caroline; Hadden, Wilbur

VERSION 1 – REVIEW

REVIEWER	Audrey Prost Institute for Global Health, University College London, UK
REVIEW RETURNED	14-Jun-2019

GENERAL COMMENTS	Thank you for the opportunity to review this interesting and useful study on differences in maternal recall between samples of mothers with infants aged 0-2 months and 0-5 months in Bihar. I recommend minor revisions: 1. The abstract's results section could be amended to include the % of indicators with different results across the core domains examined as in Table 1.2. The core domains need relabelling to avoid overlap. For example, why have ANC and birth preparedness, and then birth preparedness as separate domain? Why separate dry cord care from newborn care? These domains could be better organised and labelled.3. In the abstract and conclusion, kindly qualify the recommendation to interview samples of mothers with 0-5 months infants with a reflection on domains where this might increase/reduce coverage estimates?4. Kindly move the reflection on reasons for differences in reports of FLW visits from the conclusion to the discussion.5. In the discussion, add a reflection on differences in all indicators where significant differences were seen, including birth preparedness and health facility delivery.6. In the discussion, kindly rephrase or delete statements about this study showing there a lack of maternal recall bias. The study tested difference between two samples using maternal recall, not maternal recall vs an objective measure of events, which is what one would need to rule out maternal recall bias. What we know now is that, in this setting, whatever bias exists, it probably wouldn't be significantly different for most indicators whether we include samples of mothers with infants 0-5 months compared to 0-2 months.
---

	7. In the discussion (limitations), kindly comment on the relevance of your study for other states where LQAS already uses broader age bands. 8. The study needs copy-editing. My suggestions for this are in the attached PDF. - The reviewer provided a marked copy with additional comments. Please contact the publisher for full details.
--	---

REVIEWER	Isabel Fulcher Harvard Medical School, USA
REVIEW RETURNED	19-Jun-2019

GENERAL COMMENTS	Thank you for the opportunity to review “How well do mothers recall their own and their infant’s perinatal events? a two district study using cross-sectional stratified random sampling in Bihar, India.” This is a well-written paper that addresses an interesting question about recollection of maternal and neonatal health events. The authors wish to determine if a sample from mothers’ whose children are 0-5 months of age will capture the same information as a sample from mothers’ of children 0-2 months of age. The study is well-motivated and poised to inform data collection practice in this region. Despite the attractive features of this work, I have several concerns about the methodological aspects of this study. Major comments  1. I am unclear on why the corollary (0-2 month vs. 3-5 month) was not considered as the main research question (page 7, lines 19-26). The authors should be more explicit about why they chose to only report the results for the 0-2 months vs. 0-5 months analysis. I have two major concerns regarding the choice of comparison groups:  a. By definition, the 0-5 months group contains a 0-2 months group (let’s call this “0-2 months subset group”). One would expect the indicator estimates to approximately match between the 0-2 months subset group and the actual 0-2 months group; in fact, it would be strange if this was not the case. My main concern is that the similarities in indicator estimates between the 0-2 months group and 0-5 months group may be largely driven by the fact the 0-5 months group contains this subset – potentially “watering down” the differences. There is no discussion of this point. To obviate this concern, I think it is essential that the results comparing the 0-2 months group to the 3-5 months group are presented. b. What is the age distribution in the 0-5 months group? As this is a random sample, it should be fairly uniform (~50% should be between 3-5 months). I would like confirmation that the distributions are as expected. 2. I would appreciate discussion around sample size considerations for the study design, especially due to the fact
---

that a “meaningful difference” was defined based on a p-value cut-off of 0.05. Typically, a meaningful difference refers to a pre-specified difference in proportions between the two groups that has some substantive grounding (e.g. a difference of 5 percentage points may be appropriate in this setting). I have several further comments:

- a. A p-value greater than 0.05 does not mean that there is no difference between the two groups. As such, use of “no meaningful differences” (page 2, line 44), “no probable differences” (page 11, line 12), or “no differences” (Table 2) should be avoided. The correct interpretation is that there is not enough evidence to reject the null hypothesis that there is no difference between the groups in this study.
 - b. A supplemental table for indicators with p-values ≥ 0.05 (similar style to Table 3) should be included so that readers can view the point estimates, corresponding 95% confidence intervals, and the actual p-values. Currently, there is no transparency about what the p-values are for the “no difference” group.
 - c. For more information on the above points, a good reference: Wasserstein, Ronald L., Allen L. Schirm, and Nicole A. Lazar. "Moving to a World Beyond “ $p < 0.05$ ”." (2019): 1-19.
3. I appreciate the explicit statement of the research question on page 7 lines 10-19. However, the paper contains statements that go beyond the scope of this question and are potentially misleading. Particularly, the following statements:
- a. (page 20, lines 15-22) Rephrase “little more is learned by doubling the total sample by measuring independently 0-2 months and 3-5 months cohorts of children” to reflect the analyses that were conducted. Something like: “there was no evidence that indicator estimates changed when children 3-5 months of age are also included in the cohort”.
 - b. (page 21, lines 21-31) “mothers do not display recall errors of their perinatal health behavior in a cohort of mothers with infants 0-5 months as compared with mothers of younger infants” The study does not directly evaluate recall errors as there is no gold standard to use as a benchmark. Thus, the authors should avoid stating that there is no recall error just because there were not significant differences between these two groups.

Minor comments

1. (page 2, lines 44-46) The fact the percentages in the abstract results do not add up to 100% may confuse readers. If results are presented in this way, I would be more explicit about what these percentages represent.
2. (page 6, lines 24-25) I am unclear about what is currently being collected in practice for these various cohorts. Is information collected on the 51 indicators for both the 0-2 and 3-5 months groups? Further, are more indicators collected for the 3-5 months group? I am trying to understand if the conclusions from this study (i.e. collapse to 0-5 months group) can be readily

	implemented in the current system or if this would result in less indicators being collected in one or both current groups. 3. (page 9, lines 17-19) This statement is vague “the calculated statistics must be considered as approximations” – approximations to what? 4. (page 9, lines 35-36) Update the statement to: “However, the weighted estimates provide better point estimates for the indicators at the district level” 5. (page 9, line 56) What is meant by “readily interpretable probability scale”? The Chi-squared test statistics are not reported and are not probabilities. Is this statement referring to the p-values? Also, Supplementary File Table S2 does not exist or is not attached. 6. (page 13, Table 2) Update the third column to “No detectable difference between 0-2 and 0-5 months results ($p\text{-value} \geq 0.05$)” 7. (page 13, Table 2) Update the fourth column to “Difference between 0-2 and 0-5 months results ($p\text{-value} < 0.05$)” 8. (page 20, lines 36-54) The exclusive breastfeeding measure (#52) should not be included in the study as it is time dependent and, as the authors discuss at length, will clearly yield different results between the 0-2 and 0-5 months cohorts. Its removal would free up space to include results from the 3-5 months group.
--	--

VERSION 1 – AUTHOR RESPONSE

Reviewer 1

Major comments

1. I am unclear on why the corollary (0-2 month vs. 3-5 month) was not considered as the main research question (page 7, lines 19-26). The authors should be more explicit about why they chose to only report the results for the 0-2 months vs. 0-5 months analysis. I have two major concerns regarding the choice of comparison groups:

- a. By definition, the 0-5 months group contains a 0-2 months group (let’s call this “0-2 months subset group”). One would expect the indicator estimates to approximately match between the 0-2 months subset group and the actual 0-2 months group; in fact, it would be strange if this was not the case. My main concern is that the similarities in indicator estimates between the 0-2 months group and 0-5 months group may be largely driven by the fact the 0-5 months group contains this subset – potentially “watering down” the differences. There is no discussion of this point. To obviate this concern, I think it is essential that the results comparing the 0-2 months group to the 3-5 months group are presented.

Authors’ response:

1a. This is an interesting point. We have two responses. Firstly, the local organization that our donor asked us to investigate was that the 0-2 month results would not match the 0-5 month results. Regardless of whether there was watering down that took place the point is made that 0-5 month cohort provides similar information as 0-2. Secondly, the 0-5 month sample is not large

enough for sub-group analysis. Providing the analyses the reviewer requests would require collecting data on identical indicators with separate samples for these age groups. The data collected on the 3-5 month cohort included many fewer and different indicators than the data collected on the 0-2 month cohort. The questions included in the 3-5 month grouping have very few questions in common. We can say this in the MS.

Reviewer comment

- b. What is the age distribution in the 0-5 months group? As this is a random sample, it should be fairly uniform (~50% should be between 3-5 months). I would like confirmation that the distributions are as expected.

Authors' response:

- 1b. The distribution in the sample is 60% 0-2 months and 40% 3-5 months. (inserted in text).

Reviewer comment

2. I would appreciate discussion around sample size considerations for the study design, especially due to the fact that a "meaningful difference" was defined based on a p-value cut-off of 0.05. Typically, a meaningful difference refers to a pre-specified difference in proportions between the two groups that has some substantive grounding (e.g. a difference of 5 percentage points may be appropriate in this setting). I have several further comments:

- a. A p-value greater than 0.05 does not mean that there is no difference between the two groups. As such, use of "no meaningful differences" (page 2, line 44), "no probable differences" (page 11, line 12), or "no differences" (Table 2) should be avoided. The correct interpretation is that there is not enough evidence to reject the null hypothesis that there is no difference between the groups in this study.

Authors' response:

- 2a. The reviewer refers to a "null hypothesis". We are not performing an hypothesis test. We are making multiple comparisons of dependent measures that do not meet the assumptions for hypothesis testing. Otherwise, the text and tables are revised to refer to "measurable differences".

Reviewer comment

- b. A supplemental table for indicators with p-values ≥ 0.05 (similar style to Table 3) should be included so that readers can view the point estimates, corresponding 95% confidence intervals, and the actual p-values. Currently, there is no transparency about what the p-values are for the "no difference" group.

Authors' response:

- 2b. Thank you. See table S2.

Reviewer comment

c. For more information on the above points, a good reference: Wasserstein, Ronald L., Allen L. Schirm, and Nicole A. Lazar. "Moving to a World Beyond "p < 0.05". (2019): 1-19.

Authors' response:

2c. Thank you for this reference.

Reviewer comment

3. I appreciate the explicit statement of the research question on page 7 lines 10-19. However, the paper contains statements that go beyond the scope of this question and are potentially misleading. Particularly, the following statements:

a. (page 20, lines 15-22) Rephrase "little more is learned by doubling the total sample by measuring independently 0-2 months and 3-5 months cohorts of children" to reflect the analyses that were conducted. Something like: "there was no evidence that indicator estimates changed when children 3-5 months of age are also included in the cohort".

Authors' response:

3a. Sentence revised

Reviewer comment

b. (page 21, lines 21-31) "mothers do not display recall errors of their perinatal health behavior in a cohort of mothers with infants 0-5 months as compared with mothers of younger infants" The study does not directly evaluate recall errors as there is no gold standard to use as a benchmark. Thus, the authors should avoid stating that there is no recall error just because there were not significant differences between these two groups.

Authors' response:

3b. Thanks for this suggestion. Sentence revised

Minor comments

1. (page 2, lines 44-46) The fact the percentages in the abstract results do not add up to 100% may confuse readers. If results are presented in this way, I would be more explicit about what these percentages represent.

Authors' response:

3b. Thanks for this suggestion. Sentence revised

Reviewer comment

2. (page 6, lines 24-25) I am unclear about what is currently being collected in practice for these various cohorts. Is information collected on the 51 indicators for both the 0-2 and 3-5 months groups? Further, are more indicators collected for the 3-5 months group? I am trying to understand if the conclusions from this study (i.e. collapse to 0-5 months group) can be readily implemented in the current system or if this would result in less indicators being collected in one or both current groups.

Authors' response

2. There are 51 indicators in the 0-2 month group. There are other indicators in the 3-5 month group. Almost all those indicators are not among the 51 in the 0-2 month group. Combining the 0-2 and 3-5 month cohorts could be readily accomplished in the current system. The result would be additional indicators on 3-5 month olds that don't overlap with the 0-2 months group.

Reviewer comment

3. (page 9, lines 17-19) This statement is vague "the calculated statistics must be considered as approximations" – approximations to what?

Authors' response

3. Sentence revised.

Reviewer comment

4. (page 9, lines 35-36) Update the statement to: "However, the weighted estimates provide better point estimates *for the indicators* at the district level"

Authors' response

4. Sentence revised.

Reviewer comment

5. (page 9, line 56) What is meant by “readily interpretable probability scale”? The Chi-squared test statistics are not reported and are not probabilities. Is this statement referring to the p- values? Also, Supplementary File Table S2 does not exist or is not attached.

Authors' response

5. Thanks for catching this. The sentence is now revised. Table exists. The editor may have not sent you the supplementary file. You should request them if you wish.

Reviewer comment

6. (page 13, Table 2) Update the third column to “No *detectable* difference between 0-2 and 0-5 months results ($p\text{-value} \geq 0.05$)”

Authors' response

6. Table revised

Reviewer comment

7. (page 13, Table 2) Update the fourth column to “Difference between 0-2 and 0-5 months results ($p\text{-value} < 0.05$)”

Authors' response

7. Table revised

Reviewer comment

8. (page 20, lines 36-54) The exclusive breastfeeding measure (#52) should not be included in the study as it is time dependent and, as the authors discuss at length, will clearly yield different

results between the 0-2 and 0-5 months cohorts. Its removal would free up space to include results from the 3-5 months group.

Authors' response

8. The indicator is not deleted but the discussion is shortened. As mentioned above the 3-5 months group has different indicators to the 0-2 month group

REVIEWER 2

Thank you for the opportunity to review this interesting and useful study on differences in maternal recall between samples of mothers with infants aged 0-2 months and 0-5 months in Bihar.

I recommend minor revisions:

Reviewer comment

1. The abstract's results section could be amended to include the % of indicators with different results across the core domains examined as in Table 1.

Authors' response

The abstract is revised so the percents add to 100.

Reviewer comment

2. The core domains need relabelling to avoid overlap. For example, why have ANC and birth preparedness, and then birth preparedness as separate domain? Why separate dry cord care from newborn care? These domains could be better organised and labelled.

Authors' response

2. Thank you for the suggestion. We have relabelled, and combined the domains.

Reviewer comment

3. In the abstract and conclusion, kindly qualify the recommendation to interview samples of mothers with 0-5 months infants with a reflection on domains where this might increase/reduce coverage estimates?

Authors' response

The abstract is revised to be more specific about the domains.

Reviewer comment

4. Kindly move the reflection on reasons for differences in reports of FLW visits from the conclusion to the discussion.

Authors' response

4. We have moved the reflection section as suggested.

Reviewer comment

5. In the discussion, add a reflection on differences in all indicators where significant differences were seen, including birth preparedness and health facility delivery.

Authors' response

5. We added a discussion as you suggest.

Reviewer comment

6. In the discussion, kindly rephrase or delete statements about this study showing there a lack of maternal recall bias. The study tested difference between two samples using maternal recall, not maternal recall vs an objective measure of events, which is what one would need to rule out maternal recall bias. What we know now is that, in this setting, whatever bias exists, it probably wouldn't be significantly different for most indicators whether we include samples of mothers with infants 0-5 months compared to 0-2 months.

Authors' response

6. Statements of maternal recall revised

Reviewer comment

7. In the discussion (limitations), kindly comment on the relevance of your study for other states where LQAS already uses broader age bands.

Authors' response

7. A few words added

Reviewer comment

8. The study needs copy-editing. My suggestions for this are in the attached PDF.

Authors' response

All changes have been made in the Word document.

VERSION 2 – REVIEW

REVIEWER	Isabel Fulcher Harvard Medical School, USA
REVIEW RETURNED	21-Aug-2019

GENERAL COMMENTS	I do not believe the authors made an appreciable effort to respond to my major concerns. I remain concerned about the statistical methods used in the analyses and the conclusions drawn from the analyses. All of the below points were included in my original review, and I do not believe they have been adequately addressed. At this point, I do not feel comfortable recommending this paper for publication.  1. Despite the authors thorough response to my comment 1, the main text still maintains that “journal space limitations prevent expanding the research to assess indicators from the 3-5 month cohort vis a vis the 0-5 month cohort” (page 3, line 18), which does not match the reasoning given in their response. More importantly, the fact that the 0-5 month cohort implicitly contains a 0-2 month cohort is still not discussed as a limitation (comment 1a&b). I am now even more concerned that the “no measurable difference” conclusion may be due to the fact that the majority (60%) of this sample contains the 0-2 month cohort, which is pushing these associations towards the null. This could have been obviated by the authors at least reporting the point estimates for the 3-5 month sub-sample (from the 0-5 month sample) for my own review. 2. The authors did not respond to my concerns regarding sample size considerations for the study design (see original comment 2 for detail). However, the authors argued that they cannot include a comparison with the 3-5 month age group because the “0-5 month sample is not large enough for sub-group analysis.” As such, it seems that the authors did consider sample size for some of these analyses but the specifics remain unclear. 3. The authors response to my original comment 2a does not align with the analyses that were actually conducted and makes me question the validity (and understanding) of the statistical methods employed. The authors stated that they were “not performing an hypothesis test”; however, the entire analysis is based on hypothesis tests (Cochran-Mantel-Haenszel tests) and drawing conclusions from the resulting p-values. The authors then stated that they are conducting “multiple comparisons of dependent measures that do not meet the assumptions for hypothesis testing” – if this is the case, the authors should not utilize or draw conclusions from the CMH tests unless they attempt to account for multiple testing (which is not currently done). 4. The authors did not report the corresponding 95% confidence intervals in Table S2 (comment 2b) and did not provide a response to my comment. 5. The authors still do not explain what is meant by a “readily interpretable probability scale” in the text or provide response to my comment (page 8, line 3; minor comment 5).
--

VERSION 2 – AUTHOR RESPONSE

Reviewer Comment

I do not believe the authors made an appreciable effort to respond to my major concerns. I remain concerned about the statistical methods used in the analyses and the conclusions drawn from the analyses. All of the below points were included in my original review, and I do not believe they have been adequately addressed. At this point, I do not feel comfortable recommending this paper for publication.

Author Response

We very much regret our first response was not sufficient. On this round we make every effort to fully address your comments. We are grateful for them and hope that this response is more satisfactory. Let us begin by explaining the background to the study.

The authors analysed existing survey data to determine whether a sample of mothers of infants 0-5 months old yields results equivalent to a sample of mothers of infants 0-2 months old in the context of using lot quality assurance sampling (LQAS); there is also a sample of mothers of infants 3-5 months old. The reason for this unusual design is that an incumbent grant holder of our Donor had collected and analysed data in multiple sub-cohorts in India: 0-2 months and 3-5 months in this case. The grant holder's major premise was that the older mothers in a cohort of mothers with 0-5 months olds would not remember their perinatal behaviour. Hence, the data with mothers of 0-5 month olds would not produce concordant results to a cohort of mothers with 0-2 months olds. The donor requested the authors to assess the accuracy of this last claim as the budgetary implications were enormous for them and the Government of India. They, as we, were concerned that the incumbent NGO might be doubling their sample size needlessly, and wasting resources as a result.

Reviewer Comment

1. Despite the authors thorough response to my **comment 1**, the main text still maintains that "journal space limitations prevent expanding the research to assess indicators from the 3-5 month cohort vis a vis the 0-5 month cohort" (page 3, line 18), which does not match the reasoning given in their response. More importantly, the fact that the 0-5 month cohort implicitly contains a 0-2 month cohort is still not discussed as a limitation (**comment 1a&b**). I am now even more concerned that the "no measurable difference" conclusion may be due to the fact that the majority (60%) of this sample contains the 0- 2 month cohort, which is pushing these associations towards the null. This could have been obviated by the authors at least reporting the point estimates for the 3-5 month sub-sample (from the 0-5 month sample) for my own review.

Authors' response:

Your query is important to us and we hope our modification addresses your concerns.

- The section "Strengths and weaknesses of the study" is revised to acknowledge the reviewer's point that a direct comparison between 0-2 and 3-5 month cohorts would be informative. On the cover page we list the following limitation: "Limitation: The study is confined to assessing data in the 0-2 month sample vis a vis a 0-5 month sample of infants; a comparison between the 3-5 month sample vis a vis the 0-5 month sample would have been informative; however, insufficient overlap of variables in the 0-5 month sample and 3-5 month sample prevented us from doing so. Our study compares a 0-2 month subsample with a 3-5 month subsample as an alternative."

We have removed the phrase explaining that journal space was a limitation. Also, we have carried out additional analyses presented below.

We make four comparisons to inform an evaluation of these data. The first compares the 0-5 month sample to the 0-2 month sample and is presented in Table 2 in the main text and Tables S2-3 and further discussed in response to reviewer point 4, below. In a second comparison, indicators estimated for the 0-2 month old **subsample** are compared to the 3-5 month old **subsample** to assess the internal consistency of estimates for the entire 0-5 sample (Tables S4a-b). In the third comparison, the 0-2 **subsample** of the 0-5 sample is compared to the 0-2 month **sample** to assess the sampling variability for this age group (Tables S5a-b). Finally, in the fourth comparison, the 3-5 **subsample** of the 0-5 sample is compared to the 3-5 month **sample** (Table S6). This last comparison is limited because the 3-5 month sample collected limited data; there are only 3 indicators common to the two samples.

Table S4a has the complete results for the second comparison with point estimates for the two **subsamples** 0-2 and 3-5 of the 0-5 sample, differences and estimated confidence intervals for the estimates and differences of 104 comparisons. Fourteen indicators for which the estimated confidence interval of the difference does not include zero are listed in Table S4b. The table has 2 panels. In the top panel are 6 indicators for which a difference is also reported in Table S2. Table S2 compares the 0-2 month sample and the 0-5 month sample. For each of these indicators the reported difference is in the same direction. Two of these indicators are indicator 52, exclusive infant breast feeding, in the 2 provinces. These differences are not surprising as it is common for mothers to introduce supplemental foods as infants age. The authors speculate in their paper that the timing of the monsoon may have reduced some of the indicators for the younger infants. Of the remaining four indicators of the top panel, three of the indicator differences are negative, lower for the younger infants and might plausibly be related to a monsoon. In the bottom panel of Table S4b are 8 indicators where zero is not in the confidence interval of the difference between subsamples, which suggests difference, but the results between two full-samples in Tables S2 and S3 suggest there is no meaningful difference, except that one of them, indicator 37 in Gopalganj, does show a difference in Table S2. Of these 8, the difference between samples is in the same direction five times and in a different direction three times. For the 5 indicators where the differences are in the same direction, the absolute value of the differences between 0-2 subsample and the 3-5 subsample are larger than the absolute values of the differences between the 0-2 month sample and the 3-5 subsample. For these 5 indicators the 0-2 month **sample** is more like the 3-5 **subsample** than the 0-2 **subsample**. For the 3 indicators where the differences are in the different directions the 0-2 **subsample** closely resembles the 0-2 month **sample**; two of the estimates are different by less than 1 percent and the third by 2.4 percent.

A third comparison is in Tables S5a-b. In Table S5a the 0-2 **subsample** is compared to the 0-2 month **sample**. The expectation here would be that there are no differences because these two samples are designed to represent the same population. However, this expectation is not met; there are 10 differences where the confidence interval for the difference does not contain 0. These differences are listed in Table S5b. Nine of these 10 indicators are also among the differences in Table S3 and Table 3 in the paper. For 3 of these 9 differences the 3-5 subsample is closer to the 0-2 month sample than it is the 0-2 subsample (indicators 15, 17-2, 51); and for 3 indicators the 3-5 subsample is further from the 0-2 month sample than the 0-2 subsample (indicators 31, 33, 35). For 3 indicators the 0-2 and 3-5 subsample indicator values are nearly equal (indicators 23, 24, 26). For the final indicator in this list (indicator 17-1, the indicator for which zero is in the confidence interval of the difference between the 0-2 month sample and the 0-5 sample in Table S3) the difference between the indicator values for the 0-2 and 3-5 subsamples and the 0-2 month sample are about equal in magnitude but have opposite signs.

Results from the fourth comparison, comparing the 3-5 **subsample** of the 0-5 **sample** to the 3-5 month sample are limited, as noted above. They are in Table S6 (attached). There are

only 3 indicators in the 2 districts – six comparisons. Zero is within the confidence interval of 5 of the differences (Table S6).

We note in the paper that in making this number of comparisons one must expect that some will be large enough to be considered meaningful by chance alone. In the above analysis there are 3 comparisons of 51 indicators in 2 provinces producing 16, 14 and 10 differences and a fourth comparison with 1 difference in 6. In their paper the authors find that some of these differences are readily understood and others may be interpreted effects of monsoon rains. Post hoc interpretation is risky here; many of these differences may be noise in the data. There is evidence in the third comparison to support this position with 3 differences moving the 0-5 **sample** closer to the 0-2 month **sample**, 3 moving it away, and 4 not moving it one way or the other. Furthermore, 9 of the 10 differences in the third comparison are also differences in the first comparison, suggesting that about half the differences between the 0-2 month sample and the 0-5 sample may be due to differences between the 0-2 month **sample** and the 0-2 **subsample** of the 0-5 sample.

The second comparison, comparing the 0-2 subsample to the 3-5 subsample, provides evidence both undermining and supporting the conclusion that a 0-5 sample will provide the same answers as two samples 0-2 and 3-5. On the one hand 5 of the fourteen differences in the third comparison are also in the first, the comparison of the 0-5 sample to the 0-2 sample, suggesting that the inclusion of 3-5 month olds in the 0-5 sample might contribute to the differences. On the other hand, in 5 out of the 8 indicators that are not different between the 0-2 and 0-5 samples, the 3-5 subsample more closely matches the 0-2 month sample than does the 0-2 subsample of the 0-5 sample. That this is true in general and not only for these extreme differences is suggested by the mean absolute differences between samples which are 5.2 for the differences between the 0-2 month **sample** and 3-5 month **subsample** and 6.6 for the differences between the 0-2 month **subsample** and the 3-5 month **subsample**.

Finally, the overall result of all these comparisons is the same as that of the comparison presented in our paper: in each comparison 85% or more of the indicators show no differences between the samples. The most consistent evidence of difference is between the 0-2 **subsample** of the 0-5 sample and the 0-2 month **sample**.

We suggest adding this discussion as supplementary text to accompany the supplementary tables. It is uploaded with the revision.

2. The authors did not respond to my concerns regarding sample size considerations for the study design (see original **comment 2** for detail). However, the authors argued that they cannot include a comparison with the 3-5 month age group because the “0-5 month sample is not large enough for sub-group analysis.” As such, it seems that the authors did consider sample size for some of these analyses but the specifics remain unclear.

Authors’ response:

Thank you. The methods section of the paper is revised to be more specific about the sample size determination. The sample size is minimal and selected for classifying health service areas using LQAS principles, and does not support subgroup analyses at that level. The LQAS design was minimalist to provide the Government of Bihar with a tool which could be sustainable for health system monitoring over time, and not costly. The Block Level sample of n=19 was adequate for an LQAS classification relative to a target. This was the main point of the project’s main study. At the district level the data are aggregated and treated as stratified random samples. The corresponding sample sizes of the 0-5 sample for Aurangabad and Gobalganj are: respectively, 0-2 months, n=124 and n=160; 3-5 months, n=85 and n=106.

3. The authors response to my original **comment 2a** does not align with the analyses that were actually conducted and makes me question the validity (and understanding) of the statistical methods employed. The authors stated that they were “not performing an hypothesis test”;

however, the entire analysis is based on hypothesis tests (Cochran- Mantel-Haenszel tests) and drawing conclusions from the resulting p-values. The authors then stated that they are conducting “multiple comparisons of dependent measures that do not meet the assumptions for hypothesis testing” – if this is the case, the authors should not utilize or draw conclusions from the CMH tests unless they attempt to account for multiple testing (which is not currently done).

Authors’ response:

Thank you again for this comment. Let us try to explain our approach. The Chi square statistic is a measure of association in contingency tables. As such it lacks some of the desirable properties of other measures of association. Unlike, say, a correlation coefficient it is not bounded by -1 and 1. The range of Chi square is non-negative numbers. The upper bound is dependent upon sample size, as is the value of the statistic itself. Both increase with larger sample sizes. The value of the statistic also varies with the size of the table and degrees of freedom. These limitations make it difficult to compare one Chi square statistic to another with different samples or variables. In our paper some of these limitations are not of great concern; all of the tables are 2x2 tables and the sample sizes are approximately constant. All of the tables have the same size, 2 x 2, and one degree of freedom.

It is certainly possible to construct statistical tests with the Chi square statistic and any one of the Chi squares that we have calculated might be so interpreted. But a single test, we do not think, addresses the research question we have posed in the paper. The LQAS survey measures 51 different indicators designed to classify health care delivery centres and health behaviours of the populations served by these centres vis a vis a target coverage. These indicators are related to each other in complicated ways. The reviewer mentions the possibility of making an adjustment for the multiple comparisons. A Bonferroni adjustment would use a Chi square of about 12 as a cut-off for statistical significance. There are 2 unweighted and 1 weighted Chi squares greater than twelve. This would greatly strengthen the conclusion that a 0-5 month sample would produce the same results as the 0-2 and 3-5 month samples. However, a Bonferroni adjustment is probably too conservative and given the multiple and complex relationships among the indicators it is not at all clear what kind of adjustment might be more appropriate.

Another possible reason for looking at each comparison separately (i.e., single adjustment), is that each indicator exists to monitor a specific health service, and cannot easily be swapped for another one. However, one could consider an adjustment for multiple comparison within each health service domain. The conclusion of the paper would be strengthened again. Instead, we used the Chi squares as measures of association to select those measures with the least agreement, the largest differences, for close examination.

4. The authors did not report the corresponding 95% confidence intervals in Table S2 (**comment 2b**) and did not provide a response to my comment.

Authors’ response:

In our paper differences between indicators are assessed with Mantel-Haenszel-Cochran chi-square statistics transformed to a probability scale (Table S2). A second version of this comparison is presented in Table S3. In this table the differences between the two samples for each indicator are assessed by estimating a confidence interval for differences using standard errors for point estimates weighted to represent the district populations and standard errors calculated with a Taylor-series approximation based upon 11 PSUs in Aurangabad and 14 PSUs in Gopalganj. In this comparison, indicators estimated for the 0-2 month sample are compared to the 0-5 month sample. We also provide the 95% confidence interval for each sample’s estimate as well as for the difference – as you requested.

We identified 20 potential differences between these samples based up the chi-square probabilities in Table S2 being less than .05. These differences are listed in the paper’s Table

3. In Table S3, 16 out of these 20 differences are confirmed with the Taylor series expansion method. The four differences that are not confirmed are in indicators 27, 34, 37 and 51, all in Gopalganj. We hope this information in Tables S2 and S3 is responsive to your request.

5. The authors still do not explain what is meant by a “readily interpretable probability scale” in the text or provide response to my comment (page 8, line 3; **minor comment 5**).

Authors' response:

We apologise if we were not responsive to your request and we are sorry Table S2 was not available to you. We had uploaded it to the system. For convenience we attach it here. Table S2 has Chi-square values and the p-values. We change the text to say that in Table S2 we present Chi Sq values and corresponding p-values in Table S2, as you request.

Table S2. Indicators from mothers in Aurangabad and Gopalganj of infants aged 0-2 months compared to 0-5 months: Estimates, Cochran-Mantel-Haenszel Chi-squares, and probabilities: Bihar 2015

Domain ^a	Indicator ^b	District ^c	Unweighted				Weighted			
			Percent 0-2	Percent 0-5	Chi-square	Probability	Percent 0-2	Percent 0-5	Chi-square	Probability
1	1	1	86.6	79.9	3.6775	0.0552	85.2	77.6	4.3758	0.0365
1	1	2	85.3	85.3	0.0000	1.0000	85.0	85.4	0.0187	0.8911
1	2	1	44	40.7	0.4996	0.4797	43.0	38.3	1.0147	0.3138
1	2	2	47.7	47.0	0.0304	0.8616	47.5	48.2	0.0212	0.8843
1	3	1	84.7	80.9	1.2398	0.2655	83.8	80.4	0.9592	0.3274
1	3	2	81.2	78.6	0.5985	0.4391	80.6	77.8	0.6424	0.4229
1	4	1	60.3	63.6	0.4931	0.4826	61.1	63.1	0.1792	0.6720
1	4	2	59.4	63.2	0.8252	0.3637	57.4	61.0	0.7211	0.3958
1	5	1	42.1	45.0	0.3491	0.5546	43.2	45.7	0.2561	0.6128
1	5	2	44.7	47.7	0.4866	0.4854	43.4	46.9	0.6302	0.4273
1	6	1	59.8	59.3	0.0107	0.9177	58.9	60.1	0.0644	0.7996
1	6	2	67.7	63.5	1.0487	0.3058	66.3	63.3	0.5176	0.4719
1	7	1	42.6	39.2	0.5043	0.4776	42.0	39.5	0.2824	0.5952
1	7	2	70.3	73.3	0.6025	0.4376	69.2	72.6	0.7537	0.3853
1	8	1	44.0	45.9	0.1685	0.6815	42.7	44.8	0.2028	0.6525
1	8	2	53.4	54.1	0.0322	0.8576	52.4	54.5	0.2438	0.6215
1	9	1	45.9	42.6	0.5004	0.4793	45.8	41.7	0.7523	0.3857
1	9	2	42.9	43.6	0.0321	0.8579	42.8	43.5	0.0231	0.8792
1	10	1	67.0	59.8	2.6555	0.1032	67.2	59.6	2.9924	0.0837
1	10	2	55.3	58.3	0.5191	0.4712	54.8	58.5	0.7674	0.3810
1	40	1	11.5	17.7	3.4797	0.0621	10.6	16.0	2.8329	0.0924
1	40	2	8.6	10.2	0.3686	0.5438	8.5	9.5	0.1374	0.7109
2	11	1	98.6	98.1	0.1451	0.7033	98.5	98.1	0.0844	0.7715
2	11	2	97.0	96.6	0.0614	0.8044	96.9	97.1	0.0131	0.9089

2	12	1	68.4	66.5	0.176 3	0.6746	69.3	67.5	0.162 6	0.6868
2	12	2	79.7	79.3	0.011 6	0.9141	81.9	80.4	0.214 3	0.6434
2	13	1	3.8	2.4	0.738 4	0.3902	4.0	2.6	0.602 1	0.4378
2	13	2	3.4	5.6	1.574 5	0.2096	3.0	5.1	1.601 6	0.2057

Table S2- Continued. Indicators from mothers in Aurangabad and Gopalganj of infants aged 0-2 months compared to 0-5 months: Estimates, Cochran-Mantel-Haenszel Chi-squares, and probabilities: Bihar 2015

Domain ^a	Indicator ^b	District ^c	Unweighted				Weighted			
			Percent 0-2	Percent 0-5	Chi-square	Probability	Percent 0-2	Percent 0-5	Chi-square	Probability
2	14	1	1.4	1.9	0.1474	0.7010	1.6	2.0	0.1025	0.7489
2	14	2	1.9	3.8	1.7161	0.1902	1.7	3.6	1.8535	0.1734
3	15	1	26.3	34.0	3.1510	0.0759	25.9	33.3	2.9594	0.0854
3	15	2	47.0	56.4	4.9192	0.0266	45.7	56.0	5.8262	0.0158
3	16	1	4.3	3.8	0.0639	0.8004	4.3	3.6	0.1410	0.7073
3	16	2	10.2	13.9	1.8263	0.1766	9.7	14.3	2.8157	0.0933
3	17	1	48.8	56.5	2.5852	0.1079	48.5	55.6	2.2155	0.1366
3	17	2	53.4	62.8	4.9914	0.0255	51.8	62.6	6.5790	0.0103
3	18	1	24.9	22.5	0.3601	0.5485	26.1	23.4	0.4531	0.5009
3	18	2	39.5	45.1	2.2194	0.1363	38.2	43.7	2.1362	0.1439
3	19	1	22.6	14.4	3.4821	0.0620	23.5	14.5	4.0983	0.0429
3	19	2	45.2	40.7	1.1508	0.2834	43.4	39.4	0.6771	0.4106
3	20	1	40.7	37.8	0.4565	0.4993	44.1	39.5	1.1927	0.2748
3	20	2	51.9	57.1	1.8538	0.1733	50.0	55.9	2.3539	0.1250
3	21	1	40.1	30.4	3.6953	0.0546	43.6	31.2	6.0749	0.0137
3	21	2	58.1	52.8	1.7277	0.1887	55.6	51.6	0.8220	0.3646
3	22	1	65.1	59.3	1.5561	0.2122	65.5	59.4	1.7643	0.1841
3	22	2	62.4	68.4	2.2445	0.1341	60.2	68.0	3.6903	0.0547
3	23	1	65.0	49.0	8.3301	0.0039	62.5	47.0	7.8078	0.0052
3	23	2	68.8	63.0	1.8993	0.1682	68.0	62.5	1.5725	0.2098
4	24	1	61.2	75.1	9.2638	0.0023	62.2	75.2	8.2048	0.0042
4	24	2	65.8	74.4	4.8055	0.0284	63.5	73.0	5.6444	0.0175
4	25	1	11.5	12.9	0.2096	0.6471	11.1	12.5	0.2064	0.6496
4	25	2	20.7	18.4	0.4319	0.5111	19.3	19.1	0.0019	0.9656
4	26	1	62.7	76.6	9.4840	0.0021	63.5	76.7	8.6936	0.0032
4	26	2	68.4	75.9	3.8007	0.0512	66.9	74.3	3.5169	0.0607

4	27	1	42.1	47.8	1.440 0	0.2301	42.3	47.6	1.219 4	0.2695
4	27	2	54.5	62.4	3.490 3	0.0617	52.6	61.1	4.023 0	0.0449

Table S2- Continued. Indicators from mothers in Aurangabad and Gopalganj of infants aged 0-2 months compared to 0-5 months: Estimates, Cochran-Mantel-Haenszel Chi-squares, and probabilities: Bihar 2015

Domain ^a	Indicator ^b	District ^c	Unweighted				Weighted			
			Percent 0-2	Percent 0-5	Chi-square	Probability	Percent 0-2	Percent 0-5	Chi-square	Probability
4	28	1	8.6	8.1	0.0323	0.8575	8.2	7.6	0.0561	0.8127
4	28	2	16.2	13.9	0.5345	0.4647	15.4	13.8	0.2732	0.6012
4	29	1	43.1	48.8	1.4361	0.2308	43.3	48.5	1.1846	0.2764
4	29	2	57.5	64.7	2.9183	0.0876	55.9	63.5	3.2352	0.0721
4	30	1	29.7	35.4	1.6374	0.2007	28.7	34.2	1.5497	0.2132
4	30	2	48.1	53	1.2686	0.2600	46.9	51.4	1.0908	0.2963
4	31	1	30.1	45.9	10.9581	0.0000	29.9	44.9	9.9713	0.0016
4	31	2	47.4	53.0	1.6849	0.1943	44.5	52.7	3.6136	0.0573
4	32	1	5.3	3.8	0.4948	0.4818	5.3	3.6	0.7401	0.3896
4	32	2	9.8	9.0	0.0897	0.7646	9.1	9.9	0.0974	0.7550
4	33	1	32.5	47.8	10.1305	0.0015	32.2	46.7	9.0939	0.0026
4	33	2	50.8	56.4	1.7133	0.1906	48.2	56.2	3.5264	0.0604
4	34	1	6.2	10.0	2.0808	0.1492	6.2	9.9	1.9617	0.1613
4	34	2	13.5	9.0	2.7721	0.0959	14.4	8.9	3.9431	0.0471
4	35	1	45.0	60.3	9.7398	0.0018	44.5	59.3	9.0930	0.0026
4	35	2	64.7	69.5	1.4491	0.2287	62.5	69.0	2.5201	0.1124
5	37	1	73.2	76.6	0.6496	0.4203	72.9	74.9	0.2182	0.6404
5	37	2	80.1	86.1	3.4220	0.0643	78.9	85.5	3.9858	0.0459
5	38	1	55.5	56.5	0.0389	0.8436	54.7	55.3	0.0117	0.9139
5	38	2	52.3	61.3	4.3834	0.0363	51.2	61.6	5.8090	0.0159
5	39	1	5.7	0	3.3034	0.0691	5.4	0	3.6637	0.0556
5	39	2	10.2	8.3	0.2393	0.6247	9.1	6.6	0.3451	0.5569
5	39.5	1	1.4	0	0.0003	0.0833	1.4	0	2.8978	0.0887
5	39.5	2	1.9	1.1	0.5139	0.4735	1.8	0.9	0.7629	0.3824
6	42	1	42.6	47.4	1.0165	0.3134	43.8	47.6	0.6553	0.4182
6	42	2	41.4	46.2	1.3134	0.2518	41.8	46.3	1.1228	0.2893

6	43	1	24.9	29.7	1.326 2	0.2495	25.6	30.4	1.300 8	0.2541
6	43	2	27.1	29.7	0.496 8	0.4809	26.9	31.2	1.258 6	0.2619

Table S2- Continued. Indicators from mothers in Aurangabad and Gopalganj of infants aged 0-2 months compared to 0-5 months: Estimates, Cochran-Mantel-Haenszel Chi-squares, and probabilities: Bihar 2015

Domain ^a	Indicator ^b	District ^c	Unweighted				Weighted			
			Percent 0-2	Percent 0-5	Chi-square	Probability	Percent 0-2	Percent 0-5	Chi-square	Probability
6	44	1	9.8	12.5	1.2833	0.2573	12.2	13.0	0.3958	0.5293
6	44	2	21.1	20.1	0.0586	0.8087	21.6	18.7	0.5178	0.4718
6	45	1	7.5	10.2	0.6321	0.4266	7.8	11.1	0.8502	0.3565
6	45	2	8.2	5.6	0.4009	0.5266	5.2	8.3	0.1670	0.6828
6	46	1	55.5	58.9	0.4904	0.4837	56.6	59.2	0.2960	0.5864
6	46	2	62.8	68.0	1.6764	0.1954	61.7	66.3	1.2736	0.2591
6	47	1	34.0	32.1	0.1768	0.6741	35.9	32.8	0.4369	0.5086
6	47	2	67.3	66.5	0.0344	0.8528	66.6	68.3	0.1745	0.6762
6	48	1	65.6	67.5	0.1782	0.6729	64.9	66.6	0.1312	0.7172
6	48	2	71.1	77.8	3.2208	0.0727	70.7	78.2	3.9664	0.0464
6	49	1	62.7	59.4	0.1537	0.6950	64.5	61.2	0.1691	0.6809
6	49	2	68.1	67.2	0.0166	0.8974	67.3	66.8	0.0011	0.9735
6	50	1	49.0	55.0	1.2527	0.2630	48.4	54.7	1.3188	0.2508
6	50	2	56.8	55.9	0.0964	0.7562	56.1	55.5	0.0615	0.8042
6	51	1	81.1	44.9	14.9736	0.0000	78.0	45.4	11.7395	0.0000
6	51	2	63.3	38.9	4.0901	0.0431	63.7	41.1	3.4654	0.0627
7	52	1	70.3	59.8	5.1727	0.0229	69.2	59.7	4.1723	0.0411
7	52	2	82.3	67.3	16.2014	0.0000	82.1	68.4	13.3711	0.0000

a . 1 Antenatal care, 2 Maternal health, 3 Birth preparedness, 4 FLW Support, 5 Place of birth & attendant, 6 Neonatal Health, 7 Exclusive breastfeeding

b . For text see Table S1

c. 1 Aurangabad, 2 Gopalganj

Table S3. Weighted indicators and confidence intervals from two samples in two districts of Bihar

Indicator a	District b	0-2 Month Sample			0-5 Month Sample			Difference		
		Point estimat e	Confidence interval ^c		Point estimat e	Confidence interval ^c		Point estimat e	Confidence interval	
			Low er	Uppe r		Low er	Uppe r		Low er	Uppe r
1	1	85.2	80.2	90.2	77.6	72.6	82.6	7.6	0.1	15.2
1	2	85.0	80.5	89.6	85.4	80.9	90.0	-0.4	-6.7	5.9
2	1	43.0	36.3	49.8	38.3	31.5	45.0	4.7	-4.5	14.0
2	2	47.5	41.5	53.6	48.2	42.1	54.2	-0.6	-9.5	8.2
3	1	83.8	78.8	88.8	80.4	75.4	85.4	3.4	-3.6	10.4
3	2	80.6	75.7	85.5	77.8	72.9	82.7	2.8	-4.5	10.0
4	1	61.1	54.4	67.8	63.1	56.4	69.9	-2.0	-11.4	7.3
4	2	57.4	51.2	63.6	61.0	54.8	67.2	-3.5	-12.3	5.2
5	1	43.2	36.4	50.1	45.7	38.9	52.6	-2.5	-12.2	7.2
5	2	43.4	37.1	49.8	46.9	40.5	53.2	-3.4	-12.4	5.6
6	1	58.9	52.3	65.6	60.1	53.5	66.7	-1.2	-10.4	8.0
6	2	66.3	60.2	72.4	63.3	57.2	69.4	2.9	-5.7	11.6
7	1	42.0	35.3	48.8	39.5	32.7	46.3	2.5	-7.0	12.1
7	2	69.2	63.2	75.2	72.6	66.6	78.6	-3.4	-11.7	4.9
8	1	42.7	36.1	49.3	44.8	38.2	51.4	-2.1	-11.4	7.2
8	2	52.4	46.2	58.7	54.5	48.2	60.8	-2.1	-10.9	6.7
9	1	45.8	39.0	52.6	41.7	34.9	48.4	4.1	-5.4	13.6
9	2	42.8	36.6	49.1	43.5	37.3	49.7	-0.6	-9.5	8.2
10	1	67.2	60.9	73.6	59.6	53.2	65.9	7.6	-1.2	16.5
10	2	54.8	48.5	61.1	58.5	52.2	64.8	-3.7	-12.5	5.2
11	1	98.5	96.7	100.2	98.1	96.4	99.9	0.4	-2.2	2.9
11	2	96.9	94.8	99.1	97.1	94.9	99.3	-0.2	-3.0	2.7
12	1	69.3	63.0	75.7	67.5	61.2	73.9	1.8	-7.2	10.8
12	2	81.9	77.3	86.6	80.4	75.7	85.0	1.6	-5.2	8.4
13	1	4.0	1.2	6.7	2.6	-0.1	5.4	1.3	-2.2	4.9
13	2	3.0	1.0	5.0	5.1	3.1	7.1	-2.2	-5.5	1.2
14	1	1.6	-0.2	3.4	2.0	0.2	3.8	-0.4	-3.1	2.3
14	2	1.7	0.2	3.2	3.6	2.1	5.1	-1.9	-4.7	0.9
15	1	25.9	20.1	31.6	33.3	27.5	39.0	-7.4	-15.9	1.1
15	2	45.7	39.6	51.8	56.0	49.9	62.1	-10.3	-19.0	-1.5
16	1	4.3	1.5	7.0	3.6	0.8	6.3	0.7	-3.0	4.4
16	2	9.7	6.1	13.3	14.3	10.8	17.9	-4.7	-10.5	1.1
17	1	48.5	41.8	55.2	55.6	48.9	62.3	-7.1	-16.5	2.4
17	2	51.8	45.6	58.0	62.6	56.4	68.8	-10.8	-19.6	-2.1
18	1	26.1	20.2	32.1	23.4	17.5	29.4	2.7	-5.5	10.9
18	2	38.2	33.0	43.3	43.7	38.5	48.8	-5.5	-13.2	2.2
19	1	23.5	16.4	30.7	14.5	7.3	21.7	9.1	0.4	17.7
19	2	43.4	37.3	49.5	39.4	33.3	45.5	4.0	-4.6	12.6
20	1	44.1	38.3	50.0	39.5	33.7	45.4	4.6	-3.7	12.9
20	2	50.0	44.7	55.4	55.9	50.5	61.3	-5.9	-13.7	2.0
21	1	43.6	36.7	50.5	31.2	24.3	38.1	12.4	3.3	21.5
21	2	55.6	49.5	61.7	51.6	45.5	57.7	4.0	-4.6	12.6
22	1	65.5	59.0	72.0	59.4	52.9	65.9	6.1	-3.0	15.2
22	2	60.2	54.0	66.3	68.0	61.8	74.2	-7.8	-16.4	0.7
23	1	62.5	54.9	70.2	47.0	39.3	54.7	15.5	5.4	25.7
23	2	68.0	61.2	74.8	62.5	55.7	69.3	5.5	-3.8	14.8
24	1	62.2	55.5	68.9	75.2	68.5	81.9	-13.0	-22.0	-4.1
24	2	63.5	57.4	69.7	73.0	66.9	79.2	-9.5	-17.9	-1.1
25	1	11.1	7.0	15.1	12.5	8.4	16.5	-1.4	-7.5	4.7
25	2	19.3	14.4	24.2	19.1	14.3	24.0	0.1	-6.9	7.2
26	1	63.5	56.8	70.1	76.7	70.1	83.4	-13.2	-22.1	-4.4
26	2	66.9	60.9	73.0	74.3	68.3	80.3	-7.4	-15.7	1.0
27	1	42.3	35.6	49.1	47.6	40.8	54.4	-5.3	-14.8	4.3
27	2	52.6	46.3	58.9	61.1	54.8	67.4	-8.5	-17.2	0.3

28	1	8.2	4.6	11.8	7.6	4.0	11.2	0.6	-4.4	5.7
28	2	15.4	11.0	19.8	13.8	9.4	18.3	1.6	-4.7	7.8
29	1	43.3	36.5	50.1	48.5	41.7	55.3	-5.2	-14.8	4.3
29	2	55.9	49.7	62.2	63.5	57.2	69.8	-7.5	-16.3	1.2
30	1	28.7	22.6	34.7	34.2	28.1	40.2	-5.5	-14.3	3.2
30	2	46.9	40.5	53.3	51.4	45.0	57.8	-4.5	-13.5	4.5
31	1	29.9	23.5	36.3	44.9	38.5	51.3	-15.0	-24.4	-5.6
31	2	44.5	38.2	50.8	52.7	46.5	59.0	-8.2	-17.2	0.7
32	1	5.3	2.2	8.5	3.6	0.4	6.7	1.7	-2.3	5.8
32	2	9.1	5.7	12.6	9.9	6.4	13.4	-0.8	-6.0	4.4
33	1	32.2	25.8	38.7	46.7	40.2	53.2	-14.5	-23.9	-5.0
33	2	48.2	41.9	54.5	56.2	49.9	62.5	-8.1	-16.9	0.8
34	1	6.2	2.8	9.6	9.9	6.5	13.3	-3.7	-8.9	1.5
34	2	14.4	9.8	19.0	8.9	4.4	13.5	5.5	-0.3	11.3
35	1	44.5	37.7	51.4	59.3	52.5	66.2	-14.8	-24.4	-5.1
35	2	62.5	56.4	68.7	69.0	62.9	75.2	-6.5	-15.1	2.1
37	1	72.9	66.7	79.1	74.9	68.7	81.0	-2.0	-10.5	6.5
37	2	78.9	73.5	84.2	85.5	80.2	90.8	-6.6	-13.7	0.4
38	1	54.7	47.8	61.7	55.3	48.4	62.2	-0.5	-10.3	9.2
38	2	51.2	44.7	57.6	61.6	55.2	68.0	-10.4	-19.4	-1.5
39	1	5.4	-0.9	11.8	0.0	-6.4	6.4	5.4	-0.9	11.8
39	2	9.1	1.3	16.9	6.6	-1.2	14.4	2.5	-9.1	14.1
39.5	1	1.4	-0.2	3.0	0.0	-1.6	1.6	1.4	-0.2	3.0
39.5	2	1.8	0.2	3.4	0.9	-0.6	2.5	0.9	-1.0	2.8
40	1	10.6	6.5	14.7	16.0	12.0	20.1	-5.4	-11.7	0.8
40	2	8.5	4.9	12.2	9.5	5.8	13.1	-0.9	-5.9	4.1
42	1	43.8	36.9	50.6	47.6	40.7	54.5	-3.8	-13.3	5.6
42	2	41.8	35.6	48.1	46.3	40.1	52.6	-4.5	-13.4	4.4
43	1	25.6	19.8	31.5	30.4	24.6	36.2	-4.8	-13.3	3.7
43	2	26.9	21.5	32.4	31.2	25.7	36.6	-4.2	-12.0	3.5
44	1	12.2	7.3	17.1	13.0	8.1	17.9	-0.8	-7.7	6.1
44	2	21.6	16.3	26.8	18.7	13.4	24.0	2.8	-4.4	10.1
45	1	7.8	0.4	15.2	11.1	3.7	18.5	-3.3	-16.2	9.6
45	2	5.2	0.3	10.1	8.3	3.4	13.2	-3.1	-17.4	11.2
46	1	56.6	49.9	63.2	59.2	52.5	65.8	-2.6	-12.2	7.0
46	2	61.7	55.5	67.9	66.3	60.1	72.6	-4.6	-13.3	4.1
47	1	35.9	29.3	42.5	32.8	26.2	39.5	3.0	-6.2	12.3
47	2	66.6	60.6	72.7	68.3	62.3	74.4	-1.7	-10.1	6.7
48	1	64.9	58.5	71.4	66.6	60.1	73.0	-1.6	-10.8	7.5
48	2	70.7	64.9	76.5	78.2	72.4	84.0	-7.5	-15.2	0.2
49	1	64.5	57.1	71.8	61.2	53.9	68.5	3.3	-6.7	13.3
49	2	67.3	60.6	73.9	66.8	60.1	73.4	0.5	-8.9	9.9
50	1	48.4	40.3	56.4	54.7	46.7	62.8	-6.4	-17.4	4.7
50	2	56.1	49.0	63.2	55.5	48.4	62.6	0.7	-9.2	10.5
51	1	78.0	65.9	90.2	45.4	33.3	57.6	32.6	13.0	52.2
51	2	63.7	48.8	78.5	41.1	26.2	56.0	22.5	-2.2	47.2
52	1	69.2	62.7	75.6	59.7	53.3	66.1	9.5	0.2	18.8
52	2	82.1	77.1	87.0	68.4	63.5	73.4	13.6	6.0	21.2
a. For text see Table S1										
b. 1 Aurangabad, 2 Gopalganj										
c. Estimated with Stata command svy										

Table S4a: Point estimates and confidence intervals of indicators for subsamples of 0-2 and 3-5 month old infants in two districts: Bihar, India, 2015

Indicator ^a	District ^b	0-2 month subsample			3-5 month subsample			Difference		
		Point estimate	Confidence interval ^c		Point estimate	Confidence interval ^c		Point estimate	Confidence interval	
			Lower	Upper		Lower	Upper		Lower	Upper
1	1	79.2	71.9	86.5	75.2	66.0	84.4	4.0	-7.7	15.7
1	2	80.8	74.4	87.2	92.3	87.5	97.2	-11.5	-19.5	-3.5
2	1	36.1	27.8	44.4	41.5	31.7	51.3	-5.4	-18.2	7.4
2	2	46.5	38.1	54.8	50.6	40.3	61.0	-4.2	-17.5	9.1
3	1	81.0	75.0	87.1	79.5	71.5	87.5	1.5	-8.5	11.6
3	2	79.9	73.4	86.5	74.7	65.8	83.7	5.2	-5.9	16.3
4	1	61.8	53.1	70.4	65.2	55.0	75.3	-3.4	-16.7	10.0
4	2	62.7	54.7	70.7	58.4	48.4	68.3	4.4	-8.4	17.1
5	1	45.0	36.2	53.9	46.8	35.5	58.0	-1.7	-16.0	12.6
5	2	49.1	40.8	57.4	43.5	33.3	53.7	5.6	-7.6	18.7
6	1	60.3	51.9	68.7	59.8	49.6	70.0	0.5	-12.7	13.7
6	2	68.5	60.8	76.1	55.8	45.7	65.8	12.7	0.0	25.4
7	1	40.3	31.8	48.8	38.4	27.4	49.3	1.9	-11.9	15.8
7	2	74.5	67.2	81.9	69.8	60.3	79.3	4.7	-7.3	16.8
8	1	49.3	41.2	57.3	38.2	27.9	48.5	11.1	-2.0	24.2
8	2	60.0	52.0	67.9	46.4	36.7	56.2	13.6	1.0	26.1
9	1	39.8	31.3	48.4	44.4	33.9	54.8	-4.5	-18.0	9.0
9	2	44.7	36.6	52.8	41.7	31.5	51.9	3.0	-10.0	16.0
10	1	61.9	54.4	69.4	56.1	45.3	66.9	5.7	-7.4	18.9
10	2	63.9	56.1	71.7	50.5	40.3	60.6	13.5	0.7	26.2
11	1	98.4	96.3	100.6	97.6	94.3	100.9	0.8	-3.1	4.8
11	2	96.7	94.1	99.3	97.7	95.1	100.3	-1.0	-4.7	2.7
12	1	65.0	56.4	73.6	71.3	61.8	80.7	-6.3	-19.1	6.5
12	2	77.8	71.1	84.6	84.1	76.7	91.4	-6.2	-16.2	3.8
13	1	3.3	0.1	6.5	1.6	-1.5	4.8	1.7	-2.8	6.2
13	2	6.7	2.6	10.9	2.8	0.0	5.6	3.9	-1.1	8.9
14	1	2.2	-0.4	4.8	1.6	-1.5	4.8	0.6	-3.5	4.7
14	2	4.5	1.0	8.0	2.3	-0.4	4.9	2.2	-2.2	6.6
15	1	35.2	26.8	43.6	30.4	20.9	39.9	4.8	-7.8	17.5
15	2	56.8	48.7	64.9	54.8	44.3	65.3	2.0	-11.3	15.2
16	1	5.3	1.4	9.3	0.9	-0.8	2.6	4.5	0.2	8.7
16	2	16.6	10.2	23.0	11.0	4.8	17.1	5.7	-3.2	14.5
17	1	59.6	51.4	67.9	49.5	38.7	60.3	10.1	-3.5	23.7
17	2	64.4	56.5	72.4	59.9	49.8	70.0	4.5	-8.3	17.4
18	1	24.2	16.8	31.6	22.2	13.9	30.5	2.0	-9.1	13.1
18	2	45.9	38.1	53.8	40.3	32.0	48.6	5.7	-5.7	17.1
19	1	17.9	11.2	24.5	9.2	2.4	16.1	8.6	-0.9	18.2
19	2	40.5	32.4	48.6	37.7	29.0	46.4	2.8	-9.0	14.6
20	1	42.5	34.8	50.2	35.1	26.0	44.2	7.4	-4.5	19.4
20	2	59.4	51.7	67.0	50.7	42.3	59.1	8.6	-2.7	20.0
21	1	37.0	29.2	44.8	22.1	13.1	31.2	14.9	2.9	26.8
21	2	54.2	46.1	62.3	47.7	38.9	56.6	6.5	-5.6	18.5
22	1	62.5	54.0	70.9	54.7	44.8	64.6	7.8	-5.2	20.8
22	2	69.7	62.3	77.1	65.5	55.4	75.5	4.2	-8.2	16.7
23	1	47.5	39.1	55.9	46.1	35.0	57.3	1.4	-12.5	15.3
23	2	64.9	56.9	73.0	58.9	48.4	69.5	6.0	-7.3	19.3
24	1	75.1	67.4	82.9	75.3	66.0	84.7	-0.2	-12.4	11.9
24	2	67.6	59.7	75.5	81.0	72.7	89.3	-13.4	-24.8	-1.9
25	1	12.1	6.1	18.0	13.1	5.9	20.2	-1.0	-10.3	8.3
25	2	16.7	10.3	23.1	22.7	13.8	31.6	-6.0	-16.9	4.9
26	1	76.2	68.6	83.8	77.5	68.5	86.6	-1.3	-13.1	10.5
26	2	69.4	61.6	77.2	81.6	73.3	89.8	-12.2	-23.6	-0.8
27	1	50.4	41.5	59.3	43.4	33.0	53.9	6.9	-6.8	20.7
27	2	57.2	48.9	65.4	66.9	57.7	76.0	-9.7	-22.0	2.7

28	1	8.6	3.7	13.6	6.0	1.3	10.6	2.7	-4.1	9.5
28	2	9.8	5.1	14.5	19.7	11.4	28.1	-9.9	-19.5	-0.3
29	1	50.4	41.5	59.3	45.6	35.4	55.9	4.8	-8.8	18.4
29	2	59.2	51.1	67.4	69.7	60.5	79.0	-10.5	-22.9	1.8
30	1	32.7	24.5	41.0	36.3	25.9	46.7	-3.6	-16.9	9.7
30	2	49.2	41.0	57.4	54.7	44.3	65.1	-5.5	-18.8	7.7
31	1	42.0	33.0	51.1	49.2	39.0	59.5	-7.2	-20.9	6.5
31	2	48.6	40.3	56.8	58.9	48.8	69.0	-10.3	-23.3	2.7
32	1	3.9	0.4	7.3	3.2	-0.4	6.8	0.7	-4.3	5.7
32	2	9.4	4.2	14.5	10.7	4.7	16.8	-1.4	-9.3	6.5
33	1	43.6	34.5	52.6	51.4	41.1	61.7	-7.8	-21.6	5.9
33	2	52.9	44.8	61.0	61.1	51.1	71.2	-8.2	-21.1	4.7
34	1	7.0	2.5	11.5	14.2	7.0	21.4	-7.2	-15.7	1.2
34	2	8.6	3.8	13.4	9.5	4.0	15.0	-0.9	-8.2	6.4
35	1	57.2	48.2	66.2	62.5	52.7	72.4	-5.3	-18.7	8.0
35	2	63.2	55.1	71.4	77.6	68.6	86.6	-14.3	-26.5	-2.2
37	1	76.6	69.1	84.1	72.3	62.7	81.8	4.3	-7.8	16.4
37	2	81.6	74.9	88.3	91.3	85.8	96.8	-9.7	-18.4	-1.0
38	1	59.0	49.9	68.0	49.7	39.3	60.2	9.2	-4.6	23.1
38	2	56.1	47.8	64.4	69.7	61.2	78.2	-13.6	-25.5	-1.7
39	1	0.0	0.0	0.0	0.0	0.0	0.0	0.0	0.0	0.0
39	2	5.8	-3.1	14.7	9.6	-40.1	59.3	-3.8	-54.3	46.7
39.5	1	0.0	0.0	0.0	0.0	0.0	0.0	0.0	0.0	0.0
39.5	2	1.1	-0.4	2.6	0.7	-0.7	2.1	0.3	-1.7	2.4
40	1	17.6	10.9	24.4	13.6	7.5	19.8	4.0	-5.2	13.2
40	2	8.0	3.9	12.1	11.6	5.6	17.7	-3.7	-11.0	3.6
42	1	52.6	44.3	61.0	40.1	29.6	50.6	12.5	-0.9	25.9
42	2	47.5	39.4	55.6	44.6	34.4	54.8	2.9	-10.2	16.0
43	1	35.6	27.4	43.8	22.6	13.6	31.5	13.1	0.9	25.2
43	2	35.5	28.0	43.0	24.8	16.4	33.1	10.8	-0.5	22.0
44	1	10.1	4.2	16.0	17.6	9.5	25.7	-7.5	-17.6	2.5
44	2	20.6	14.1	27.0	16.2	8.7	23.8	4.3	-5.6	14.3
45	1	9.0	-4.7	22.7	13.7	-0.7	28.1	-4.7	-24.6	15.2
45	2	6.6	-9.9	23.2	14.4	14.4	14.4	-7.7	-24.3	8.8
46	1	63.4	54.6	72.2	52.8	41.5	64.0	10.7	-3.6	24.9
46	2	68.9	61.1	76.7	62.5	52.6	72.4	6.4	-6.2	19.0
47	1	33.0	24.7	41.4	32.5	22.2	42.9	0.5	-12.8	13.8
47	2	70.4	62.8	77.9	65.3	55.9	74.8	5.1	-7.1	17.2
48	1	68.8	60.6	77.0	63.2	52.5	73.8	5.7	-7.8	19.1
48	2	76.2	69.3	83.0	81.3	73.9	88.6	-5.1	-15.2	5.0
49	1	66.5	57.9	75.1	52.7	41.0	64.3	13.8	-0.6	28.3
49	2	68.6	59.9	77.3	64.4	53.8	75.0	4.2	-9.5	17.9
50	1	51.7	42.1	61.4	59.5	47.6	71.5	-7.8	-23.1	7.0
50	2	55.2	46.3	64.1	55.8	45.1	66.6	-0.7	-14.6	13.3
51	1	33.2	14.6	51.7	61.0	36.7	85.4	-27.9	-58.5	2.7
51	2	40.6	17.5	63.7	43.0	-6.7	92.7	-2.4	-57.2	52.4
52	1	73.0	65.1	80.9	39.7	28.6	50.7	33.3	19.8	46.9
52	2	83.8	78.2	89.3	45.8	35.6	55.9	38.0	26.4	49.5

a. For text see Table S1

b. 1 Aurangabad, 2 Gopalganj

c. Estimated with Stata command svy

Table S4b: Indicators in Table S4 where zero is not in the estimated confidence interval of difference between 0-2 and 3-5 month subsamples

Indicator	District ^a	Point estimate		Difference	95% Confidence interval ^b	
		0-2 month	3-5 month		Lower	Upper
In Table S4a and in Table S2						
21	1	37.0	22.1	14.9	2.9	26.8
24	2	67.6	81.0	-13.4	-24.8	-1.9
37	2	81.6	91.3	-9.7	-18.4	-1.0
38	2	56.1	69.7	-13.6	-25.5	-1.7
52	1	73.0	39.7	33.3	19.8	46.9
52	2	83.8	45.8	38.0	26.4	49.5
In Table S4a but not in Table S2						
1	2	80.8	92.3	-11.5	-19.5	-3.5
8	2	60.0	46.4	13.6	1.0	26.1
10	2	63.9	50.5	13.5	0.7	26.2
16	1	5.3	0.9	4.5	0.2	8.7
26	2	69.4	81.6	-12.2	-23.6	-0.8
28	2	9.8	19.7	-9.9	-19.5	-0.3
35	2	63.2	77.6	-14.3	-26.5	-2.2
43	1	35.6	22.6	13.1	0.9	25.2

a. 1 = Aurangabad; 2 = Gopalganj

b. Calculated from standard errors for point estimates estimated with Stata command svy

Table S5a: Point estimates and confidence intervals of indicators for a sample 0-2 month old infants and a subsample of 0-2 month old infants from a sample of 0-5 month old infants in two districts: Bihar, India, 2015

Indicator ^a	District ^b	0-2 month subsample			0-2 month sample			Difference		
		Point estimate	Confidence interval ^c		Point estimate	Confidence interval ^c		Point estimate	Confidence interval	
			Lower	Upper		Lower	Upper		Lower	Upper
1	1	79.2	71.9	86.5	85.2	80.2	90.2	-6.0	-14.8	2.8
1	2	80.8	74.4	87.2	85.0	80.5	89.6	-4.2	-12.1	3.6
2	1	36.1	27.8	44.4	43.0	36.3	49.8	-6.9	-17.6	3.8
2	2	46.5	38.1	54.8	47.5	41.5	53.6	-1.1	-11.3	9.2
3	1	81.0	75.0	87.1	83.8	78.8	88.8	-2.8	-10.7	5.1
3	2	79.9	73.4	86.5	80.6	75.7	85.5	-0.7	-8.8	7.5
4	1	61.8	53.1	70.4	61.1	54.4	67.8	0.7	-10.3	11.6
4	2	62.7	54.7	70.7	57.4	51.2	63.6	5.3	-4.8	15.4
5	1	45.0	36.2	53.9	43.2	36.4	50.1	1.8	-9.4	13.0
5	2	49.1	40.8	57.4	43.4	37.1	49.8	5.7	-4.8	16.2
6	1	60.3	51.9	68.7	58.9	52.3	65.6	1.4	-9.3	12.1
6	2	68.5	60.8	76.1	66.3	60.2	72.4	2.2	-7.6	12.0
7	1	40.3	31.8	48.8	42.0	35.3	48.8	-1.7	-12.6	9.2
7	2	74.5	67.2	81.9	69.2	63.2	75.2	5.3	-4.2	14.8
8	1	49.3	41.2	57.3	42.7	36.1	49.3	6.5	-3.9	16.9
8	2	60.0	52.0	67.9	52.4	46.2	58.7	7.6	-2.6	17.1
9	1	39.8	31.3	48.4	45.8	39.0	52.6	-5.9	-16.8	5.0
9	2	44.7	36.6	52.8	42.8	36.6	49.1	1.8	-8.3	12.0
10	1	61.9	54.4	69.4	67.2	60.9	73.6	-5.3	-15.2	4.5
10	2	63.9	56.1	71.7	54.8	48.5	61.1	9.1	-0.9	19.1
11	1	98.4	96.3	100.6	98.5	96.7	100.2	-0.0	-2.8	2.8
11	2	96.7	94.1	99.3	96.9	94.8	99.1	-0.2	-3.7	3.2
12	1	65.0	56.4	73.6	69.3	63.0	75.7	-4.3	-15.0	6.4
12	2	77.8	71.1	84.6	81.9	77.3	86.6	-4.1	-12.3	4.1
13	1	3.3	0.1	6.5	4.0	1.2	6.7	-0.6	-4.9	3.6
13	2	6.7	2.6	10.9	3.0	1.0	5.0	3.7	-0.9	8.3
14	1	2.2	-0.4	4.8	1.6	-0.2	3.4	0.7	-2.5	3.8
14	2	4.5	1.0	8.0	1.7	0.2	3.2	2.8	-1.1	6.7
15	1	35.2	26.8	43.6	25.9	20.1	31.6	9.3	-0.8	19.5
15	2	56.8	48.7	64.9	45.7	39.6	51.8	11.1	0.9	21.2
16	1	5.3	1.4	9.3	4.3	1.5	7.0	1.1	-3.7	5.9
16	2	16.6	10.2	23.0	9.7	6.1	13.3	7.0	-0.4	14.3
17	1	59.6	51.4	67.9	48.5	41.8	55.2	11.1	0.5	21.7
17	2	64.4	56.5	72.4	51.8	45.6	58.0	12.7	2.6	22.7
18	1	24.2	16.8	31.6	26.1	20.2	32.1	-1.9	-11.4	7.6
18	2	45.9	38.1	53.8	38.2	33.0	43.3	7.8	-1.6	17.1
19	1	17.9	11.2	24.5	23.5	16.4	30.7	-5.7	-15.5	4.1
19	2	40.5	32.4	48.6	43.4	37.3	49.5	-2.9	-13.0	7.3
20	1	42.5	34.8	50.2	44.1	38.3	50.0	-1.6	-11.3	8.0
20	2	59.4	51.7	67.0	50.0	44.7	55.4	9.3	0.0	18.7
21	1	37.0	29.2	44.8	43.6	36.7	50.5	-6.6	-17.0	3.8
21	2	54.2	46.1	62.3	55.6	49.5	61.7	-1.4	-11.6	8.7
22	1	62.5	54.0	70.9	65.5	59.0	72.0	-3.0	-13.7	7.7
22	2	69.7	62.3	77.1	60.2	54.0	66.3	9.5	-0.1	19.2
23	1	47.5	39.1	55.9	62.5	54.9	70.2	-15.0	-26.4	-3.6
23	2	64.9	56.9	73.0	68.0	61.2	74.8	-3.1	-13.7	7.4
24	1	75.1	67.4	82.9	62.2	55.5	68.9	12.9	2.7	23.2
24	2	67.6	59.7	75.5	63.5	57.4	69.7	4.1	-5.9	14.1
25	1	12.1	6.1	18.0	11.1	7.0	15.1	1.0	-6.2	8.2
25	2	16.7	10.3	23.1	19.3	14.4	24.2	-2.6	-10.6	5.5
26	1	76.2	68.6	83.8	63.5	56.8	70.1	12.7	2.6	22.8
26	2	69.4	61.6	77.2	66.9	60.9	73.0	2.4	-7.4	12.3
27	1	50.4	41.5	59.3	42.3	35.6	49.1	8.1	-3.1	19.3
27	2	57.2	48.9	65.4	52.6	46.3	58.9	4.6	-5.8	15.0

28	1	8.6	3.7	13.6	8.2	4.6	11.8	0.5	-5.7	6.6
28	2	9.8	5.1	14.5	15.4	11.0	19.8	-5.6	-12.0	0.8
29	1	50.4	41.5	59.3	43.3	36.5	50.1	7.1	-4.1	18.3
29	2	59.2	51.1	67.4	55.9	49.7	62.2	3.3	-7.0	13.6
30	1	32.7	24.5	41.0	28.7	22.6	34.7	4.1	-6.1	14.3
30	2	49.2	41.0	57.4	46.9	40.5	53.3	2.3	-8.1	12.7
31	1	42.0	33.0	51.1	29.9	23.5	36.3	12.1	1.1	23.2
31	2	48.6	40.3	56.8	44.5	38.2	50.8	4.1	-6.3	14.3
32	1	3.9	0.4	7.3	5.3	2.2	8.5	-1.5	-6.2	3.2
32	2	9.4	4.2	14.5	9.1	5.7	12.6	0.2	-6.0	6.4
33	1	43.6	34.5	52.6	32.2	25.8	38.7	11.3	0.2	22.5
33	2	52.9	44.8	61.0	48.2	41.9	54.5	4.7	-5.5	15.0
34	1	7.0	2.5	11.5	6.2	2.8	9.6	0.8	-4.8	6.4
34	2	8.6	3.8	13.4	14.4	9.8	19.0	-5.8	-12.5	0.8
35	1	57.2	48.2	66.2	44.5	37.7	51.4	12.7	1.3	24.0
35	2	63.2	55.1	71.4	62.5	56.4	68.7	0.7	-9.5	10.9
37	1	76.6	69.1	84.1	72.9	66.7	79.1	3.7	-6.0	13.4
37	2	81.6	74.9	88.3	78.9	73.5	84.2	2.7	-5.8	11.3
38	1	59.0	49.9	68.0	54.7	47.8	61.7	4.2	-7.2	15.6
38	2	56.1	47.8	64.4	51.2	44.7	57.6	4.9	-5.6	15.4
39	1	0.0	0.0	0.0	5.4	-0.9	11.8	-5.4	-11.8	0.9
39	2	5.8	-3.1	14.7	9.1	1.3	16.9	-3.3	-15.1	8.5
40	1	0.0	0.0	0.0	1.4	-0.2	3.0	-1.4	-3.0	0.2
40	2	1.1	-0.4	2.6	1.8	0.2	3.4	-0.7	-2.9	1.4
40	1	17.6	10.9	24.4	10.6	6.5	14.7	7.0	-0.9	14.9
40	2	8.0	3.9	12.1	8.5	4.9	12.2	-0.6	-6.0	4.9
42	1	52.6	44.3	61.0	43.8	36.9	50.6	8.9	-2.0	19.7
42	2	47.5	39.4	55.6	41.8	35.6	48.1	5.7	-4.6	15.9
43	1	35.6	27.4	43.8	25.6	19.8	31.5	10.0	-0.1	20.1
43	2	35.5	28.0	43.0	26.9	21.5	32.4	8.6	-0.7	17.8
44	1	10.1	4.2	16.0	12.2	7.3	17.1	-2.1	-9.8	5.6
44	2	20.6	14.1	27.0	21.6	16.3	26.8	-1.0	-9.3	7.3
45	1	9.0	-4.7	22.7	7.8	0.4	15.2	1.2	-14.3	16.8
45	2	6.6	-9.9	23.2	5.2	0.3	10.1	1.4	-15.8	18.7
46	1	63.4	54.6	72.2	56.6	49.9	63.2	6.9	-4.2	17.9
46	2	68.9	61.1	76.7	61.7	55.5	67.9	7.2	-2.8	17.2
47	1	33.0	24.7	41.4	35.9	29.3	42.5	-2.8	-13.5	7.8
47	2	70.4	62.8	77.9	66.6	60.6	72.7	3.7	-6.0	13.4
48	1	68.8	60.6	77.0	64.9	58.5	71.4	3.9	-6.5	14.3
48	2	76.2	69.3	83.0	70.7	64.9	76.5	5.4	-3.6	14.4
49	1	66.5	57.9	75.1	64.5	57.1	71.8	2.1	-9.2	13.3
49	2	68.6	59.9	77.3	67.3	60.6	73.9	1.3	-9.7	12.3
50	1	51.7	42.1	61.4	48.4	40.3	56.4	3.4	-9.2	15.9
50	2	55.2	46.3	64.1	56.1	49.0	63.2	-0.9	-12.3	10.4
51	1	33.2	14.6	51.7	78.0	65.9	90.2	-44.9	-67.0	-22.7
51	2	40.6	17.5	63.7	63.7	48.8	78.5	-23.0	-50.5	4.5
52	1	73.0	65.1	80.9	69.2	62.7	75.6	3.8	-6.3	14.0
52	2	83.8	78.2	89.3	82.1	77.1	87.0	1.7	-5.7	9.1

a. For text see Table S1

b. 1 Aurangabad, 2 Gopalganj

c. Estimated with Stata command svy

Table S5b: Point estimates of weighted indicators and differences between a sample of mothers of 0-2 month old infants and a subsample of mothers of 0-2 month old infants from a sample of mothers of 0-5 month old infants in two districts of Bihar, India

Indicator	District ^a	Point estimates		Difference		
		Subsample	Sample	Estimate	Confidence interval ^b	
					Lower	Upper
15 Proportion of mothers (home + institutional delivery) of infants (0-2/0-5) months who planned transportation to health facility in their last pregnancy	2	56.8	45.7	11.1	0.9	21.1
17 Proportion of mothers (home + institutional delivery) of infants (0-2/0-5) months who have identified persons who would take care of the baby immediately after birth	1	59.6	48.5	11.1	0.5	21.1
	2	64.4	51.8	12.7	2.6	22.5
23 Proportion of mothers who planned for institutional delivery of infants (0-2/0-5) months identified person to accompany her during the delivery	1	47.5	62.5	-15.0	-26.4	-3.6
24 Proportion of mothers of infants (0-2/0-5) months who were visited by ASHA at least once during their last pregnancy	1	75.1	62.2	12.9	2.7	23.6
26 Proportion of mothers of infants (0-2/0-5) months who were visited by FLWs at least once during their last pregnancy	1	76.2	63.5	12.7	2.6	22.9
31 Proportion of mothers of infants (0-2/0-5) months who were visited home by ASHA within 24 hours of last delivery	1	42.0	29.9	12.1	1.1	23.1
33 Proportion of mothers of infants (0-2/0-5) months who were visited home by any FLW within 24 hours of last delivery	1	43.6	32.2	11.3	0.2	22.3
35 Proportion of mothers of infants (0-2/0-5) months who were visited home by any FLW within first week of last delivery	1	57.2	44.5	12.7	1.3	24.3
51 Proportion of infants aged (0-2/0-5) months who were delivered at home continued with dry cord care	1	33.2	78.0	-44.9	-67.0	-22.1
c.	1 = Aurangabad; 2 = Gopalganj					
d.	Calculated from standard errors for point estimates estimated with Stata command svy					

Table S6: Point estimates and confidence intervals of indicators for a sample of 3-5 month infants and subsample of 3-5 month infants in two districts: Bihar, India, 2015

Indicator ^a	District ^b	3-5 month subsample			3-5 month sample			Difference		
		Point estimate	Confidence interval ^c		Point estimate	Confidence interval ^c		Point estimate	Confidence interval	
			Lower	Upper		Lower	Upper		Lower	Upper
37	1	72.3	62.7	81.8	70.4	64.2	76.5	1.9	13.3	-9.5
37	2	91.3	85.8	96.8	86.2	81.8	90.6	5.1	12.2	-2.0
38	1	49.7	39.3	60.2	45.8	38.9	52.6	4.0	16.4	-8.5
38	2	69.7	61.2	78.2	64.4	58.3	70.5	5.4	15.8	-5.1
52	1	39.7	28.6	50.7	65.3	58.8	71.8	-25.6	-12.8	-38.4
52	2	45.8	35.6	55.9	55.2	49.1	61.2	-9.4	2.4	-21.2

a. For text see Table S1

b. 1 Aurangabad, 2 Gopalganj

c. Estimated with Stata command svy